# Beyond the Known: Decision Making with Counterfactual Reasoning Decision Transformer

## Abstract

Decision Transformers (DT) play a crucial role in modern reinforcement learning, leveraging offline datasets to achieve impressive results across various domains. However, DT requires high-quality, comprehensive data to perform optimally. In real-world applications, the lack of training data and the scarcity of optimal behaviours make training on offline datasets challenging, as suboptimal data can hinder performance. To address this, we propose the Counterfactual Reasoning Decision Transformer (CRDT), a novel framework inspired by counterfactual reasoning. CRDT enhances DT's ability to reason beyond known data by generating and utilizing counterfactual experiences, enabling improved decision-making in unseen scenarios. Experiments across Atari and D4RL benchmarks, including scenarios with limited data and altered dynamics, demonstrate that CRDT outperforms conventional DT approaches. Additionally, reasoning counterfactually allows the DT agent to obtain stitching abilities, combining suboptimal trajectories, without architectural modifications. These results highlight the potential of counterfactual reasoning to enhance reinforcement learning agents' performance and generalization capabilities.

## 1 Introduction

In the pursuit of achieving artificial general intelligence (AGI), reinforcement learning (RL) has been a widely adopted approach. Conventional RL methods have shown impressive success in training AI agents to perform tasks across various domains, such as gaming (Mnih et al., 2015; Silver et al., 2017) and robotic manipulation (Van Hoof et al., 2015). When referring to conventional RL approaches, we mean methods that train agents to discover an optimal policy that maximizes returns (Sutton, 2018), either through value function estimation (Watkins & Dayan, 1992) or policy gradient derivation (Sutton et al., 1999). However, more recent advances, such as Decision Transformers (DT) (Chen et al., 2021), introduce a paradigm shift by leveraging supervised learning on offline RL datasets, offering a more practical and scalable alternative to the online learning traditionally required in RL. This shift highlights the growing importance of supervised learning on offline RL approaches, which can be more efficient and convenient in environments where data collection is expensive or impractical (Srivastava et al., 2019; Chen et al., 2021; Janner et al., 2021).

In its original form, the DT agent is trained to maximize the likelihood of actions conditioned on past experiences (Chen et al., 2021). Numerous follow-up studies have tried to improve DT, such as through online fine-tuning (Zheng et al., 2022), pre-training (Xie et al., 2023), or improving its stitching capabilities (Wu et al., 2024; Zhuang et al., 2024). These works have shown that DT techniques can match or even outperform state-of-the-art conventional RL approaches on certain tasks. However, these improvements focus solely on maximizing the use of available data, raising the question: What if the optimal data is underrepresented in the given dataset? This scenario is illustrated in Fig. 1 of a toy navigation environment, wherein the blue (good) trajectories are underrepresented compared to the green (bad) trajectories. The traditional DT by Chen et al. (2021) is expected to underperform in this environment because it simply maximizes the likelihood of the training data, which can be problematic when optimal data is lacking. Additionally, it lacks effective stitching capabilities—the ability to combine suboptimal trajectories (refer to Appendix. A for an

explanation of the stitching behaviour). This leads us to a key question: *Can we improve DT's performance by enabling the agent to reason about what lies beyond the known?*

Our Counterfactual Reasoning Decision Transformer (CRDT) approach is inspired by the potential outcome framework, specifically, the ability to reason counterfactually (Neyman, 1923; Rubin, 1978). The core intuition behind CRDT is that by reasoning about hypothetical, better outcomes, the agent can deepen its understanding of the environment and the relationships between states, actions, and rewards, ultimately improving its generalizability. This mirrors how humans imagine alternative scenarios and outcomes from past experiences to inform better decisions in the future.

The CRDT framework has three key steps. The first step involves training the agent to reason counterfactual. We introduce two models: the Treatment model $\mathcal{T}$ and the Outcome model $\mathcal{O}$. The model $\mathcal{T}$ is trained to estimate the conditional distribution of actions given the historical experiences, i.e., the probability of selecting actions based on past trajectories. This differs from the original DT, which directly predicts the action itself rather than modeling the underlying distribution. The model $\mathcal{O}$ is trained to predict the future state and return as outcomes of taking an action. Once these two models are trained using the given offline dataset, we proceed to the second step. We aim to utilize the action selection probabilities and the inferred outcomes to generate counterfactual experiences. Unlike prior approaches that generate counterfactual data simply by perturbing the actions or states (Pitis et al., 2022; Sun et al., 2023; Zhao et al., 2024; Sun et al., 2024) with small noise, we argue that an action should be considered as counterfactual if only it has a low probability of being selected. We employ a mechanism known as *Counterfactual Action Selection* mechanism to identify such actions. However, extreme counterfactual actions may introduce excessive noise or lead to states that are not beneficial for the agent's learning. To mitigate this, we implement a mechanism called *Counterfactual Action Filtering* to eliminate irrelevant actions. The actions that pass the filtering process will be used as inputs for the Outcome model, which gives us the outcomes of these actions. In the final step, we integrate these counterfactual experiences with the offline dataset to train the underlying DT agent. Fig. 1(c) provides an overview of our CRDT framework.[1]

Our empirical experiments in continuous action space environments, including locomotion, ant and maze benchmarks, small datasets, and modified environment settings, and discrete action space environments like Atari, show that our framework improves the performance of the underlying DT agent. Moreover, we demonstrate that under CRDT, the DT agent attains the "stitching" ability without needing to modify the underlying architecture. To summarize, our key contributions are:

1. We propose the CRDT framework, which enables agents to reason counterfactually, allowing them to explore alternative outcomes and generalize to novel scenarios.

2. Through extensive experiments, we demonstrate that CRDT consistently enhances the performance of the underlying DT agent and provides it with the ability to stitch trajectories. This improvement is observed across various conditions, including standard settings, smaller datasets, and modified environments.

## 2 PRELIMINARIES

### 2.1 OFFLINE REINFORCEMENT LEARNING AND DECISION TRANSFORMER

We consider learning in a Markov decision process (MDP) represented by the tuple $(S, A, r, P, \gamma, \rho_0)$, where $S$ is the state space, $A$ is the action space, reward function $r : S \times A \to \mathbb{R}$, $\gamma$ is the discount factor, and the initial distribution $\rho_0$. At each timestep $t$, the agent observes a state $s_t \in S$, takes an action $a_t \in A$ and receives a reward $r_t = R(s_t, a_t)$. The transition to the next state $s_{t+1} \in S$ follows the probability transition function $P(s_{t+1} \mid s_t, a_t)$. The goal of reinforcement learning is to find a policy $\pi(a|s)$ that can maximize the expected return $\mathbb{E}_{\pi, P, \rho_0} \left[ \sum_{t=0}^{\infty} \gamma^t R(s_t, a_t) \right]$.

In **offline RL**, the agent is not allowed to interact with the environment until test time (Levine et al., 2020). Instead, it is given a static dataset $\mathcal{D}_{\text{env}} = \{(s_0^{(i)}, a_0^{(i)}, r_0^{(i)}, s_1^{(i)}, \ldots, s_t^{(i)}, a_t^{(i)}, r_t^{(i)}, \ldots)\}_{i=1}^{N}$, collected from one or more behaviour policies $\pi_\beta$, to learn from. Generally, learning the optimal policy from a static dataset is challenging or even impossible (Kidambi et al., 2020). Consequently, the objective is to create algorithms that reduce sub-optimality to the greatest extent possible.

---

[1]Source code will be made available upon publication.

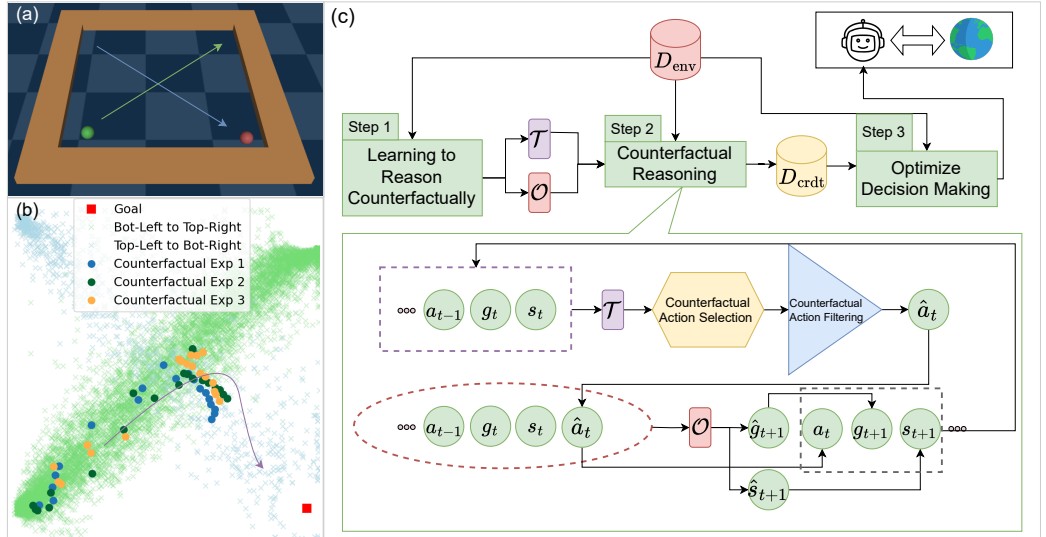

Figure 1: **(a)**: A toy environment where the goal of the agent is to move from the green circle position to the red circle position given that data is biased toward moving from bottom-left to top-right (green trajectory) over top-left to bottom-right (blue trajectory). When using traditional DT, the agent will most likely follow the green trajectory and fail to reach the goal. **(b)**: The empirical result of the counterfactual reasoning process following CRDT on the toy environment. At the crossing between green and blue trajectories, notice that turning right yields a higher potential outcome/return, CRDT generates counterfactual experience accordingly. As shown by the bold yellow, blue, and green dots, none of the counterfactual experiences followed the green trajectory after the crossing point; they all show a clear right turn. Training DT with these counterfactual experiences improved the overall performance (refer to Sect. 4.4.2 for performance results). **(c) Top**: The CRDT framework follows three steps: first, learning to reason counterfactually with the CRDT agent; second, perform counterfactual reasoning to generate counterfactual experiences; and third, use these experiences to improve decision-making. **Bottom**: A single step in the iterative counterfactual reasoning process of a trajectory. The outcomes of one-step reasoning are the counterfactual action $\hat{a}_t$, the next state $\hat{s}_{t+1}$ and returns-to-go $\hat{g}_{t+1}$ will replace the original values $a_t, s_{t+1}, g_{t+1}$ and the generated data will be used in next iteration.

**Decision Transformer** (DT) (Chen et al., 2021) is a pioneering work that frames RL as a sequential modeling problem. The authors introduce a transformer-based agent, denoted as $\mathcal{M}$ with trainable parameters $\delta$, to tackle offline RL environments. While substantial research has built upon this work (see Sect. 5 for a comprehensive review), DT, in its original form, applies minimal modifications to the underlying transformer architecture (Vaswani, 2017). Similar to traditional offline RL approaches, the agent $\mathcal{M}$ in DT is given an offline dataset $\mathcal{D}_{\text{env}}$, which contains multiple trajectories. Each trajectory consists of sequences of states, actions, and rewards. However, rather than simply using past rewards from $\mathcal{D}_{\text{env}}$ as input into $\mathcal{M}$, the authors introduce returns-to-go, denoted as $g_t$ and computed as $g_t = \sum_{t'=t}^{T} r_{t'}$. The agent $\mathcal{M}$ is fed this returns-to-go $g_t$ instead of the immediate reward $r_t$, allowing it to predict actions based on future desired returns. In Chen et al. (2021), a trajectory $\tau^{(i)}$ is represented as: $\tau^{(i)} = (g_1^{(i)}, s_1^{(i)}, a_1^{(i)}, \ldots, g_T^{(i)}, s_T^{(i)}, a_T^{(i)})$. Agent $\mathcal{M}$ with parameter $\delta$ is trained on a next action prediction task. This involves using the experience $h_t = (g_1, s_1, a_1, ..., g_t, s_t, a_t)$, returns-to-go $g_{t+1}$ and state $s_{t+1}$ as inputs and the next action $a_{t+1}$ as output. This can be formalized as:

$$p(a_{t+1} \mid h_t, s_{t+1}, g_{t+1}; \delta) = \mathcal{M}(h_t, s_{t+1}, g_{t+1}; \delta), \qquad (1)$$

for discrete action space. And:

$$a_{t+1} = \mathcal{M}(h_t, s_{t+1}, g_{t+1}; \delta). \qquad (2)$$

for continuous action space. This action prediction ability is then utilized during the inference and evaluation phases on downstream RL tasks. In addition to the aforementioned process, the authors investigated the potential benefits of integrating additional tasks to predict the next state and returns-to-go into the agent's training to enhance its understanding of the environment's structure, however, it was concluded that such methods do not improve the agent's performance (Chen et al., 2021). Further, they suggested that this "would be an interesting study for future research" (Chen et al., 2021). Our method, while not explicitly incorporating such predictions, demonstrates an alternative approach that can effectively use these predictions to improve the agent's performance.

## 2.2 POTENTIAL OUTCOME AND COUNTERFACTUAL REASONING

Our work is inspired by the potential outcomes framework (Neyman, 1923; Rubin, 1978) and its extension to time-varying treatments and outcomes (Robins & Hernan, 2008). The potential outcomes framework is a key approach in causal inference that defines and estimates causal effects by considering the potential outcomes for each variable under different treatment conditions (Robins & Hernan, 2008). Counterfactual reasoning involves imagining what might have happened under alternative conditions or scenarios that did not occur (Pearl & Mackenzie, 2018). Under the potential outcome framework, at each timestep $t \in \{1, ..., T\}$, we observe time-varying covariates $X_t$, treatments $A_t$, and the outcomes $Y_{t+1}$. The treatment $A_t$ influences the outcome $Y_{t+1}$, and all $X_t$, $A_t$, and $Y_{t+1}$ affect future treatment. A history at timestep $t$ is denoted as $\bar{H}_t = \{\bar{X}_t, \bar{A}_{t-1}, \bar{Y}_t\}$, where $\bar{X}_t = (X_1, \ldots, X_t)$, $\bar{Y}_t = (Y_1, \ldots, Y_t)$, and $\bar{A}_{t-1} = (A_1, \ldots, A_{t-1})$. The estimated potential outcome for a trajectory of treatment $\bar{a}_t = (a_t, ..., a_{t+\xi-1})$ is expressed as $\mathbb{E}[Y_{t+\xi}(\bar{a}_{t:t+\xi-1}) \mid \bar{H}_t]$ where $\xi \geq 1$ is the treatment horizon for $\xi$ steps prediction.

Mapping to this paper, the time-varying covariates correspond to the agent's past observations and the returns-to-go it has received. The treatment corresponds to the action taken, and the outcome is the subsequent observation and future returns. A counterfactual treatment refers to an action the agent could have taken but did not. Therefore, for each timestep $t$, we aim to estimate the outcome of counterfactual action $\hat{a}_t$ or $\mathbb{E}[\hat{s}_{t+1}, \hat{g}_{t+1} \mid \hat{h}_t]$, where $\hat{s}_{t+1}$ and $\hat{g}_{t+1}$ denote the counterfactual state and returns-to-go corresponding to taking the counterfactual action $\hat{a}_t$. $\hat{h}_t$ is the new historical experience $(g_1, s_1, a_1, \ldots, g_t, s_t, \hat{a}_t)$, given that we have taken a counterfactual action $\hat{a}_t$ that is different from the original action $a_t$ in the dataset $D_{\text{env}}$.

Our framework follows the three standard assumptions: (1) consistency, (2) sequential ignorability, and (3) sequential overlap ensuring that the counterfactual outcomes over time are identifiable from the factual observational data $\mathcal{D}_{\text{env}}$ (see Appendix. B).

# 3 METHODOLOGY

This section introduces the Counterfactual Reasoning Decision Transformer framework, our approach to empowering the DT agent with counterfactual reasoning capability.[2] The framework follows three steps: first, we train the Treatment and Outcome Networks to reason counterfactually; then, we use these two networks to generate counterfactual experiences and add these to a buffer $D_{\text{crdt}}$; and finally, we train the underlying agent with these new experiences.

## 3.1 LEARNING TO REASON COUNTERFACTUALLY

As mentioned in Sect. 2.2, counterfactual reasoning involves estimating how outcomes would differ under unobserved treatments (Pearl & Mackenzie, 2018). This process is often broken down into learning the selection probability of the agent's treatment and learning the outcomes of the treatments. This means that we must be able to estimate the probability of selecting actions $a_t$, at timestep $t$, given historical experiences $h_{t-1} = (g_1, s_1, a_1, \ldots, g_{t-1}, s_{t-1}, a_{t-1})$, the current outcome state $s_t$, and returns-to-go $g_t$. Knowing the distribution enables exploration of counterfactual actions $\hat{a}_t$ (actions with low selection probability). By using these counterfactual actions as new treatment, we can estimate their corresponding outcomes, the next state $\hat{s}_{t+1}$ and the next returns-to-go $\hat{g}_{t+1}$. To address these steps, we introduce two separate transformer models: the Treatment

---

[2]From this point forward, if needed we will use the notations $a_t^*, s_t^*, g_t^*$ for the factual values and notations $a_t, s_t, g_t$ for the predicted values. $\hat{a}_t, \hat{s}_t, \hat{g}_t$ will be used to denote counterfactual related values.

model ($\mathcal{T}$) and the Outcome model ($\mathcal{O}$). The model $\mathcal{T}$, parameterized by $\theta$, learns the probability of selecting treatments (i.e., the agent's action). The model $\mathcal{O}$, with parameters $\eta$, estimates the outcomes of actions. Together, these models enable the agent to reason counterfactually, by learning the probability of selecting actions and the potential outcomes of unchosen actions.

**Treatment Model Training.** We want to use the Treatment model $\mathcal{T}$ to estimate the probability of selecting a specific action. In discrete action space environment, this can be formalized as:

$$p(a_t \mid h_{t-1}, s_t, g_t; \theta) = \mathcal{T}(h_{t-1}, s_t, g_t; \theta). \tag{3}$$

The model can be trained using a cross-entropy (CE) loss:

$$\mathcal{L}_{\mathcal{T}(\theta)} = -\frac{1}{N} \sum_{i=1}^{N} a_t^{*(i)} \log \left( p(a_t^{(i)} \mid h_{t-1}^{(i)}, s_t^{(i)}, g_t^{(i)}; \delta) \right). \tag{4}$$

where $a_t^{*(i)}$ is the encoded true label for the action of the $i$-th instance of N samples, and $p(a_t^{(i)} \mid h_{t-1}^{(i)}, s_t^{(i)}, g_t^{(i)}; \delta)$ is the predicted probability of the action $a_t^{(i)}$. In environments with a continuous action space, we assume that actions follow a Gaussian distribution and estimate its mean and variance using a neural network (an assumption that is often made in continuous treatment potential outcome research (Robins et al., 2000; Zhu et al., 2015; Bahadori et al., 2022)), thus, $a_t \sim \mathcal{N}(\mu_t, \sigma_t^2)$, where $\mu_t, \sigma_t^2 = \mathcal{T}(h_{t-1}, s_t, g_t; \theta)$. The model $\mathcal{T}$ is trained to minimize:

$$\mathcal{L}_{\mathcal{T}(\theta)} = \frac{1}{N} \sum_{i=1}^{N} \left( \frac{(a_t^{*(i)} - \mu_t^{(i)})^2}{2\sigma_t^{2(i)}} + \frac{1}{2} \log(2\pi\sigma_t^{2(i)}) \right). \tag{5}$$

**Outcome Model Training.** To predict outcome of taking an action, the $\mathcal{O}$ model is trained to minimize the loss between predicted state $s_{t+1}$ and returns-to-go $g_{t+1}$ and their factual values. This objective can be achieved using the Mean Squared Error (MSE) loss. This can be formalized as:

$$s_{t+1}, g_{t+1} = \mathcal{O}(h_t; \eta), \tag{6}$$

$$\mathcal{L}_{\mathcal{O}(\eta)} = \frac{1}{N} \sum_{i=1}^{N} \left( \|s_{t+1}^{*(i)} - s_{t+1}^{(i)}\|^2 + \|g_{t+1}^{*(i)} - g_{t+1}^{(i)}\|^2 \right). \tag{7}$$

Here, $s_{t+1}^{(i)}$ and $g_{t+1}^{(i)}$ are two different output heads of the outcome model with input trajectory $h_t = (g_1, s_1, a_1, ..., g_t, s_t, a_t)$. **The training procedures of model $\mathcal{T}$ and $\mathcal{O}$ are detailed in the Algorithm. 1 in Appendix. C**.

## 3.2 COUNTERFACTUAL REASONING WITH CRDT

This section describes the agent's iterative counterfactual reasoning process. At each timestep $t$, the model $\mathcal{T}$ is provided with the input sequence $(g_1, s_1, a_1, \ldots, g_{t-1}, s_{t-1}, a_{t-1}, g_t, s_t)$ to compute the action distribution. Using this distribution, a counterfactual action $\hat{a}_t$ is drawn according to the Counterfactual Action Selection. Next, the model $\mathcal{O}$ is used to generate the counterfactual state $\hat{s}_{t+1}$ and returns-to-go $\hat{g}_{t+1}$. The trajectory is then updated with the counterfactual experience, forming the new input $(g_1, s_1, a_1, \ldots, g_t, s_t, \hat{a}_t, \hat{g}_{t+1}, \hat{s}_{t+1})$ for the next iteration. Counterfactual reasoning for a trajectory is deemed successful if the iterative process proceeds to the end of the trajectory without violating the Counterfactual Action Filtering mechanism. Successful reasoning trajectories are added to the counterfactual experience buffer, denoted as $D_{\text{crdt}}$, if the number of experiences in $D_{\text{crdt}}$ is less than a hyperparameter $n_e$. **The counterfactual reasoning process is detailed in the Algorithm. 2 in Appendix. D**.

**Counterfactual Action Selection.** Our goal is to sample $n_a$ actions that can be classified as counterfactual actions, which will be passed to the filtering process. Rather than just adding small noise, we aim to identify counterfactual actions as outliers, thereby, encouraging the exploration of less

supported outcomes. The method for selecting a counterfactual action differs based on whether the action space is discrete or continuous. In a discrete action space, as the output of the Treatment model is the probability of the action, we can simply select all actions whose probability of being selected is less than a threshold $\gamma$. On the other hand, for continuous action spaces, we draw inspiration from the maximum of Gaussian random variables, as discussed in Kamath (2015), to derive our bound to identify counterfactual actions. Specifically, the upper bound of the expectation of the maximum of Gaussian random variables is used. Applying to action $a_t$, this is written as:

$$\mathbb{E}\left[\max(a_t)\right] \leq \mu_t + \sqrt{2}\sigma_t \sqrt{\ln(n_{enc})}. \tag{8}$$

Here, $n_{enc}$ denotes the number of times the model has encountered an input $(h_t, s_{t+1}, g_{t+1})$. This bound indicates the expected range for the action, and any action that exceeds this bound is considered a counterfactual action. Based on this, we derive the formula to search for potential actions in the counterfactual action set (detailed in Appendix. D):

$$a_t^{(j)} = \mu_t - \Phi^{-1}\left(0.08 - j \cdot \beta\right)\sigma_t\sqrt{\ln(n_{enc})}, \quad \text{for } j = 0, 1, \ldots, n_a. \tag{9}$$

where $\beta$ is the step size and $j$ indicates the index of the $j$-th action from the total $n_a$ sampled actions. $\Phi^{-1}$ is the quantile function of the standard normal distribution. When $j = 0$, $\Phi^{-1}\left(0.08 - j \cdot \beta\right) = \Phi^{-1}\left(0.08\right) \approx -\sqrt{2}$, thus, Eq. 9 is approximately equal to the RHS of Eq. 8. By using Eq. 9, we ensure that at each time step $t$, we can explore a diverse range of candidate counterfactual actions.

**Counterfactual Action Filtering.** This mechanism is proposed to filter counterfactual actions that are not beneficial to the agent. For each candidate action, we generate subsequent outcomes using $\mathcal{O}$ to construct candidate counterfactual trajectories. The trajectories are then filtered based on 2 criteria: (1) high accumulated return and (2) high prediction confidence. The reason for sampling high return actions is because DT techniques improve with higher return data (Bhargava et al., 2024; Zhao et al., 2024), aligning with our approach to introduce counterfactual experiences that can lead to better outcomes. Therefore, we look for actions that resulted in the lowest counterfactual returns-to-go (equivalent to higher return), $\hat{g}_{t+1}$, lower than returns-to-go $g_{t+1}$ in the offline dataset $D_{env}$.

Regarding the second criterion, we introduce an uncertainty estimator function to determine low prediction confidence states and exclude actions that lead to these states, therefore stopping and discarding the counterfactual trajectory if the uncertainty is too high. There are multiple ways to implement such an estimator. In our framework, the model $\mathcal{O}$ is trained with dropout regularization layers. This allows us to run multiple forward passes through the model, with the dropout layer activated, to check the uncertainty of the output state. The output of $m$ forward passes, at timestep $t$, is the matrix of state predictions, $\mathbf{S}_{t+1} = \begin{bmatrix} s_{t+1}^{(1)} & s_{t+1}^{(2)} & \cdots & s_{t+1}^{(m)} \end{bmatrix}$. $\mathbf{S}_{t+1} \in \mathbb{R}^{m \times d}$, where $d$ is the dimension of each prediction. We denote $\text{Var}(\mathbf{S}_k)$, where $k$ is a timestep, as the function that calculates the maximum variance across all dimensions $j'$ of $s_k$, where $j' = 1, 2, \ldots, d$. This can be obtained from the covariance matrix of $\mathbf{S}_k$ (detailed in Appendix. D.2). The maximum variance across all dimensions is used as the variance of the predictions and the uncertainty value. Our uncertainty filtering mechanism, checking the accumulated maximum variance, can be written as:

$$U^\alpha(\mathbf{S}_{t+1}) = \begin{cases} \textbf{TRUE} & \text{(Unfamiliar)}, & \text{if } \sum_{k=t_0}^{t+1} \text{Var}(\mathbf{S}_k) > \alpha, \\ \textbf{FALSE} & \text{(Familiar)}, & \text{otherwise}. \end{cases} \tag{10}$$

Here, $\sum_{k=t_0}^{t+1}\left(\text{Var}(\mathbf{S}_k)\right)$ is the accumulated maximum variances of state prediction from a timestep $t_0$ that we start the reasoning process to current checking timestep $t + 1$. The function $U^\alpha(\mathbf{S}_{t+1})$ returns **TRUE** if the state $s_{t+1}$ is unfamiliar. If the uncertainty is low, we will run a final forward pass through the model, with the dropout layer deactivated, to get the deterministic state and returns-to-go output. This helps avoid noise accumulation and supports beneficial counterfactual reasoning.

### 3.3 OPTIMIZING DECISION-MAKING WITH COUNTERFACTUAL EXPERIENCE

In this section, we describe how our counterfactual reasoning capability has been applied to improve the agent's decision-making. To demonstrate the effectiveness, we have selected the original DT

model introduced by Chen et al. (2021) as the main backbone for the experiment. The learning agent in this paper, denoted as $\mathcal{M}$, is trained following Eq. 1 to minimize either CE loss for discrete action space environments or Eq. 2 with MSE loss for continuous action space environments. For discrete action space, the loss function is defined as:

$$\mathcal{L}_{\mathcal{M}(\delta)} = -\frac{1}{N} \sum_{i=1}^{N} a_{t+1}^{*(i)} \log \left( p(a_{t+1}^{(i)} \mid h_t^{(i)}, s_{t+1}^{(i)}, g_{t+1}^{(i)}; \delta) \right). \tag{11}$$

where $a_{t+1}^{*(i)}$ is the encoded true label for the action of the $i$-th instance of N samples, and $p(a_{t+1}^{(i)} \mid h_t^{(i)}, s_{t+1}^{(i)}, g_{t+1}^{(i)}; \delta)$ is the probability output from the model. For continuous actions, the loss is:

$$\mathcal{L}_{\mathcal{M}(\delta)} = \frac{1}{N} \sum_{i=1}^{N} \left( \|a_{t+1}^{*(i)} - a_{t+1}^{(i)}\|^2 \right). \tag{12}$$

At each training step, we sample equal batches of trajectories from both the environment dataset $D_{\text{env}}$ and the counterfactual experience buffer $D_{\text{crdt}}$. The agent $\mathcal{M}$ is trained on both data sources, with the total loss calculated as the combination of the two losses $\mathcal{L}_{\mathcal{M}(\delta)} = \mathcal{L}_{\mathcal{M}(\delta)}^{\text{env}} + \mathcal{L}_{\mathcal{M}(\delta)}^{\text{crdt}}$. **The training procedure of agent $\mathcal{M}$ is described in Algorithm. 3 in Appendix. E**. We also explore potential combinations of our framework with other DT techniques in Appendix. F.8.

## 4 EXPERIMENTS

We conduct our experiments on both continuous action space environments (Locomotion, Ant, and Maze2d from the D4RL benchmark (Fu et al., 2020)) and discrete action space environments (Atari (Bellemare et al., 2013)) to address several key research questions: *Does CRDT enhance the underlying DT algorithm comparing to other variants in standard benchmarks* (Sect. 4.1 and Sect. 4.3)? *Can CRDT improve DT's generalizability when trained on a limited $D_{\text{env}}$ dataset or modified evaluating environments* (Sect. 4.2 and Appendix. F.3)? *What is the impact of selecting out-of-distribution actions* (Sect. 4.4.1)? *Can CRDT enable DT to stitch trajectories without altering the underlying backbone architecture* (Sect. 4.4.2)?

We compare our method with several baselines, including conventional RL and sequential modeling techniques. For implementation details, refer to Appendix. F.1. Conventional methods include Behavior Cloning (BC) (Pomerleau, 1988), model-free offline methods, such as Conservative Q-Learning (CQL) (Kumar et al., 2020) and Implicit Q-Learning, (IQL) (Kostrikov et al., 2021b) and model-based offline methods, such as MOPO (Yu et al., 2020) and MOReL (Kidambi et al., 2020). Sequential modeling baselines include simple backbone DT (Chen et al., 2021), Elastic Decision Transformer (EDT) (Wu et al., 2024) and state-of-the-art Reinformer (REINF) (Zhuang et al., 2024).

### 4.1 DOES CRDT ENHANCE THE DT IN CONTINUOUS ACTION SPACE ENVIRONMENTS?

We present the experimental results of CRDT compared to other baselines on the standard Locomotion benchmark and the Ant task from the D4RL dataset. As shown in Table. 1, CRDT consistently enhances the performance of the simple backbone DT model across all datasets. Specifically, it achieves an average of 3.5% improvement on the Locomotion tasks and a 2.7% improvement on the Ant task. Notably, the largest gain occurs on the walker2d-mediumrlay dataset, with a significant 16.1% increase (please refer to Appendix. F.10 for a visualization of how CRDT's counterfactual action distribution differs from the original data distribution). On average, CRDT is also the best-performing method, outperforming all other methods on the Locomotion task, and demonstrates results comparable to the state-of-the-art reinforcement learning approach, IQL, and sequential modeling approach, REINF, on the Ant task.

### 4.2 CAN CRDT IMPROVE DT'S PERFORMANCES GIVEN LIMITED TRAINING DATASET?

To evaluate the generalizability improvements of CRDT over DT, we conducted experiments using only a limited subset of the $D_{\text{env}}$ dataset. The experiments were carried out on Locomotion and

Table 1: Performance comparison on Locomotion and Ant tasks. We rort the results over 5 seeds. For each seed, evaluation is conducted over 100 episodes. The best result is shown in **bold**, and the second-best is in *italic*. ♠ denotes the best Sequence Modeling approaches.

| Dataset | Traditional Methods | | | | | Sequence Modeling Methods | | | |
|---|---|---|---|---|---|---|---|---|---|
| | BC | CQL | IQL | MOPO | MOReL | DT | EDT | REINF | **CRDT** |
| halfcheetah-med | 42.6 | *44.0* | **47.4** | 42.3 | 42.1 | 42.6 | 42.1 | 42.8♠ | 42.82±2.3♠ |
| halfcheetah-med-rep | 36.6 | *45.5* | 42.4 | **53.1** | 40.2 | 36.1 | 37.8 | 38.3♠ | 38.03±2.5 |
| halfcheetah-med-exp | 55.2 | *91.6* | 86.7 | 63.3 | 53.3 | 90.2 | 82.0 | 91.2 | **96.4±2.3♠** |
| hopper-med | 52.9 | 58.5 | 66.3 | 28.0 | **95.4** | 67.9 | 59.6 | *75.2♠* | 67.94±1.5 |
| hopper-med-rep | 18.1 | **95.0** | *94.7* | 67.5 | 93.6 | 85.0 | 76.1 | 84.2 | 85.54±3.2♠ |
| hopper-med-exp | 52.5 | 105.4 | 91.5 | 23.7 | *108.7* | 108.7 | 92.4 | 107.6 | **110.37±0.1♠** |
| walker2d-med | 75.3 | 72.5 | 78.3 | 17.8 | 77.8 | 75.9 | 66.4 | *77.9* | **78.95±0.9♠** |
| walker2d-med-rep | 26.0 | **77.2** | *73.9* | 39.0 | 49.8 | 62.1 | 58.1 | 72.1 | 72.28±0.1♠ |
| walker2d-med-exp | 107.5 | *108.8* | **109.6** | 44.6 | 95.6 | 108.5 | 106.9 | 108.7 | 109.05±0.6♠ |
| **Locomotion** | 466.7 | 698.5 | 692.6 | 378.0 | 656.5 | 677.0 | 621.4 | 698.0 | **701.38±1.5** |
| ant-med-rep | - | - | **92.0** | - | - | 90.0 | 85.0 | *91.6♠* | 91.02±8.8 |
| ant-med | - | - | *93.9* | - | - | 91.5 | 90.7 | 92.7 | **95.84±8.3♠** |
| **Ant** | - | - | *186* | - | - | 181.5 | 175.7 | 184.3 | **186.86±8.5** |

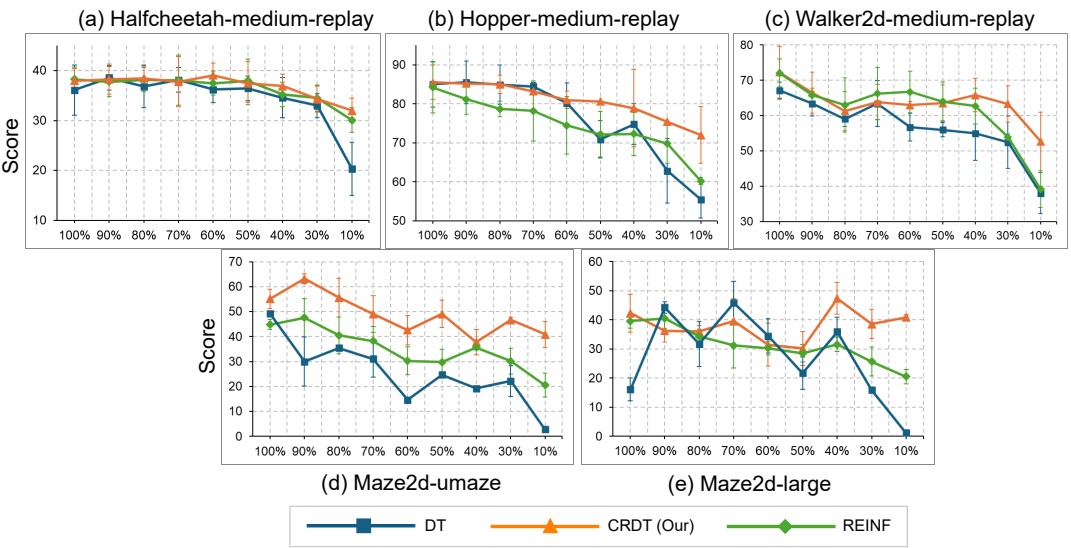

Figure 2: Performance comparison on limited subset of $D_{\text{env}}$ dataset. We report the results over 5 seeds. For each seed, evaluation is conducted over 100 episodes. The X-axis represents the percentage of the dataset used in the experiment.

Maze2d (more challenging environments as they required the ability to stitch suboptimal trajectories (Zhuang et al., 2024)) tasks. We compared CRDT's performance against the backbone DT model and Reinformer, the second-best sequence modeling method according to Table. 1. The results of this experiment are shown in Fig. 2. According to this figure, our method experiences the smallest performance degradation in this setting. In the Halfcheetah and Hopper environments, where all three methods exhibit similar performance at 100% dataset size, our method demonstrates only about a 15% performance drop when trained on 10% of the dataset. In contrast, both REINF and DT degrade by over 21%, with extreme cases approaching a 40% decline. On the Maze tasks, CRDT performances drop approximately 25% on the umaze and 3% on the large dataset. In contrast, the simple backbone DT approach cannot learn these environments (performance drop more than 90%) and REINF performance drops approximately 45% given only 10% of the dataset.

### 4.3 DOES CRDT ENHANCE THE DT IN DISCRETE ACTION SPACE ENVIRONMENTS?

We also conducted experiments on four Atari games, which features discrete action spaces and more complex observation spaces. These are the environments that were used in Chen et al. (2021). The normalized scores are shown in Table. 2 (raw scores can be found in Table. 5 Appendix. F.4). Given

Table 2: Performance comparison (scores normalized according to Table. 6 Appendix. F.4) on Atari games (1% DQN-replay dataset). We report the results over 3 seeds. For each seed, evaluation is conducted over 10 episodes. The best result is shown in **bold**. ♠ indicates games in which CRDT improves the backbone DT approach.

| Game | BC | DT | CRDT (Ours) |
|---|---|---|---|
| Breakout | 138.9±54.6 | 198.6±1.8 | **248.9±58.9**♠ |
| Qbert | **17.4±13.4** | 7.2±0.2 | 7.5±0.6♠ |
| Pong | 85.2±78.3 | **140.2±63.6** | 102.2±67.6 |
| Seaquest | 2.1±0.2 | 5.7±6.3 | **7.4±0.5**♠ |
| Average | 60.9±36.6 | 87.9±17.9 | **91.5±31.9**♠ |

Table 3: Performance comparison with different action selection methods on walker2d-med-rep.

| Variations | Score |
|---|---|
| DT | 62.1±2.2 |
| W/o comparing $g$ | 67.4±2.1 |
| W/o $U^\alpha(\mathbf{S}_k)$ | 69.6±2.8 |
| $a$ | 68.4±3.45 |
| $a$ + noise $\epsilon$ | 69.3±4.4 |
| CRDT (Ours) | **72.3±0.1** |

the increased difficulty of the observation space, we anticipated that CRDT might not always out-perform DT, as it could introduce higher levels of noise, even with mechanisms in place to prevent noise accumulation. Nevertheless, CRDT improved DT in 3 out of the 4 games (highest improvement of 25% on Breakout), though there was a performance drop in one. We believe that for these complex environments, a larger neural network (we use the same network for $\mathcal{M}$ model for $\mathcal{T}$ and $\mathcal{O}$ models) could lead to greater performance gains.

## 4.4 ABLATION STUDY

### 4.4.1 COMPARING CRDT WITH VARYING ACTION SELECTION METHODS

We conduct an ablation study on the two mechanisms that define our method: Counterfactual Action Filtering and Counterfactual Action Selection, using the walker2d-medium-replay dataset. In Table. 3, we compare the performance of the full CRDT against several variations: the version that does not compare the returns-to-go (denoted as W/o comparing $g$), the version that does not utilize the uncertainty quantifier $U^\alpha(\mathbf{S}_k)$, the variation that simply samples an action $a$ without considering whether $a$ is an out-of-distribution action, and the variation that samples an action $a + \epsilon$ as the counterfactual action, where $\epsilon$ is random Gaussian noise sampled from the range [0.01, 0.05]. The results from this experiment show that simply adding data will improve the performance of backbone DT, however, the improvement is less significant than when our framework CRDT is used. Full CRDT improves the performance by 16%, while the closet variations, do not utilize $U^\alpha(\mathbf{S}_k)$ and sample action $a + \epsilon$, achieving only 12.0% and 11.6%.

### 4.4.2 CAN CRDT ENABLE DT TO STITCH TRAJECTORIES?

Table. 7 in Appendix F.7 presents the results of the experiment conducted in the environment shown in Fig. 1. In this environment, all states, apart from the goal, receive a reward of 0. Reaching the goal state receives a reward of +1. We expect that, if traditional DT is used, the agent would struggle to learn this environment due to the lack of stitching ability and the lack of optimal data. The results in the table support our expectations, indicating that the traditional DT achieves only around a 40% success rate, whereas our CRDT approach achieves nearly 90%. Although our approach has not been designed to achieve stitching ability during training, such as in Wu et al. (2024) and Zhuang et al. (2024), our agent acquires this ability by training on data that has already been stitched together through the process of counterfactual reasoning and the generation of higher-return counterfactual experiences. This also explains the performance in the Ant dataset in Table. 1 and the Maze2d dataset (especially when the data is small) in Fig. 2, both of which require trajectory stitching.

## 5 RELATED WORK

### 5.1 OFFLINE REINFORCEMENT LEARNING AND SEQUENCE MODELING

Offline RL (Levine et al., 2020) refers to the task of learning policies from a static dataset $D_{env}$ of pre-collected trajectories. It has found successful applications in robotic manipulation (Kalashnikov et al., 2018; Mandlekar et al., 2020) and healthcare (Wang et al., 2018; Tang et al., 2022). Traditional methods used to solve offline RL can be classified into model-free offline RL and model-based RL approaches. Model-free methods aim to constrain the learned policy close to the behaviour policy (Levine et al., 2020), through techniques such as learning conservative Q-values (Kumar et al., 2020; Xie et al., 2021; Kostrikov et al., 2021a), applying uncertainty quantification to the predicted Q-values (Agarwal et al., 2020; Levine et al., 2020), and incorporating regularization based on importance sampling (Sutton et al., 2016; Liu et al., 2019). Other methods include imposing state and action constraints using various distance metrics, such as imitation loss (Fujimoto et al., 2019), MSE constraint (Fujimoto & Gu, 2021), or KL divergence (Liu et al., 2022). Model-based offline RL methods (Yu et al., 2020; Kidambi et al., 2020; Yu et al., 2021; Rigter et al., 2022), involve learning the dynamic model of the environment, then, generating rollouts from the model to optimize the policy. Our method is more aligned with model-based approaches, as we use a model to generate counterfactual samples. However, the difference is that we only sample low selection action.

Before the development of Decision Transformer (DT), upside-down reinforcement learning (Srivastava et al., 2019; Schmidhuber, 2019) applied supervised learning techniques to address RL tasks. In 2021, Chen et al. (2021) introduced Decision Transformer (DT) and the concept of incorporating returns into the sequential modeling process to predict optimal actions. In the same year, Trajectory Transformer (TT) (Janner et al., 2021) presented a different approach to representing input trajectories. Inspired by both DT and TT, numerous methods have since been proposed to enhance performance, focusing on areas such as architecture (Kim et al., 2023; Bar-David et al., 2023), pretraining (Xie et al., 2023), online fine-tuning (Zheng et al., 2022), dynamic programming (Yamagata et al., 2023), and trajectory stitching (Wu et al., 2024; Zhuang et al., 2024). However, up to our knowledge, there has been no work that seeks to integrate counterfactual reasoning with DT.

### 5.2 COUNTERFACTUAL REASONING IN CONVENTIONAL REINFORCEMENT LEARNING

Several methods have explored the application of counterfactual reasoning in RL (Buesing et al., 2018; Oberst & Sontag, 2019; Pitis et al., 2020; Mesnard et al., 2020; Pitis et al., 2022; Killian et al., 2022) and imitation learning (IL)(Sun et al., 2023). While these approaches leverage counterfactual reasoning, they are not directly comparable to our method. The key distinction lies in their reliance on the Structural Causal Model (SCM) framework (Pearl & Mackenzie, 2018). These works necessitate either a pre-defined causal graph or the learning of such a graph from data. In contrast, our approach is rooted in the Potential Outcomes (PO) framework (Rubin, 1978; Robins & Hernan, 2008), which focuses on estimating the effects of interventions without the need for a specified causal graph. This allows us to avoid the need to learn the causal graph. Our approach aligns more closely with works that estimate counterfactual outcomes for treatments in sequential data (Melnychuk et al., 2022; Frauen et al., 2023; Wang et al., 2018; Li et al., 2020); all of which adopt the PO framework. However, the main contribution of our work lies in integrating these estimated outcomes to enhance the decision-making of the underlying DT agent (refer to Appendix G for the relation of CRDT to causal inference and counterfactual reasoning).

## 6 DISCUSSION

In this paper, we present the CRDT framework, which integrates counterfactual reasoning with DT. Our experiments show that CRDT improves DT and its variants on standard benchmarks and in scenarios with small datasets while generalizing to modified evaluation environments. Additionally, the agent achieves trajectory stitching without architectural changes. However, training separate Transformer models adds complexity. Future work could explore combining these models, as they share inputs, or training in an iterative manner using generated counterfactual samples as training data, though careful consideration is needed.

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

## A    STICHING BEHAVIOR IN SEQUENTIAL MODELING

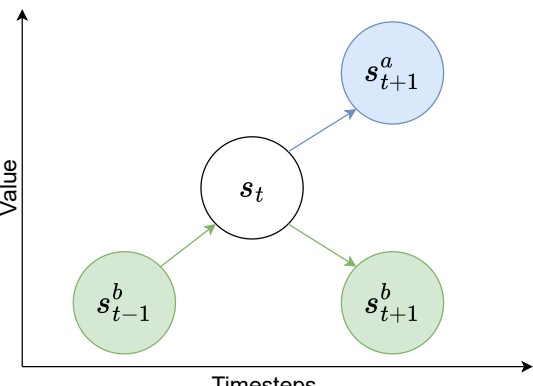

Figure 3: Given two trajectories $\left(s_{t-1}^a, s_t, s_{t+1}^a\right), \left(s_{t-1}^b, s_t, s_{t+1}^b\right)$. We want our agent to be able to start from state $s_{t-1}^b$, however, can reach state $s_{t+1}^a$

Trajectory stitching is an ability that has received great attention lately in offline RL, specifically, in sequential modeling. It has been proven that traditional sequential modeling approaches, such as Chen et al. (2021), lack the ability to stitch suboptimal trajectories to form optimal trajectories (Kumar et al., 2020; Zhuang et al., 2024; Wu et al., 2024). An example of this is given in the toy environment provided in Fig. 1(a) and Fig. 3. Let's consider the scenario of two trajectories $\left(s_{t-1}^a, s_t, s_{t+1}^a\right), \left(s_{t-1}^b, s_t, s_{t+1}^b\right)$ where $\left(s_{t-1}^a, s_t, s_{t+1}^a\right)$ is sampled from the set of blue trajectories (good trajectories that lead to the goal) and $\left(s_{t-1}^b, s_t, s_{t+1}^b\right)$ is sampled from the set of green trajectories (bad trajectories that do not lead to the goal). We anticipate that a sequence model trained on these trajectories will likely follow the subsequent states in a way that aligns with the provided trajectories. This means that the agent starting from the bottom-left potentially follows the green trajectories to the top-right of the maze and will not reach the goal. We want, however, to stitch these trajectories together, meaning that we want our agent to be able to start from $s_{t-1}^b$ but end up being in $s_{t+1}^a$.

The explanation for why traditional DT does not have stitching ability arises from the agent's training conditions. Specifically, when using traditional DT, the prediction of the next state-action pair is conditioned on an initial target return ($g_0$). If $g_0$ is set to 0, the ball will smoothly follow the green trajectory, as this is the more common data and the returns-to-go at the crossroad (the point where the green and blue trajectories intersect) are still equal to 0. On the other hand, if conditioned on a return of 1, the ball is likely to take a random action because $g_0 = 1$ represents an out-of-distribution (OOD) returns-to-go from the bottom-left corner of the maze. In both cases, the ball fails to reach the goal. Previous works, such as Wu et al. (2024) and Zhuang et al. (2024) address this problem by modifying the training condition of the DT agent. Our approach, on the other hand, addresses this problem by generating better trajectories based on the idea of potential outcome, thus, guiding the agent to reach the goal.

## B   POTENTIAL OUTCOME FRAMEWORK AND ASSUMPTIONS FOR CAUSAL IDENTIFICATION

We build upon the potential outcomes framework (Neyman, 1923; Rubin, 1978) and its extension to time-varying treatments and outcomes (Robins & Hernan, 2008). In order to identify the counterfactual outcome distribution over time, the following three standard assumptions for the data-generating process are required:

**Assumption A.1 (Consistency)**: If $\bar{A}_t = \bar{a}_t$ is a fixed sequence of treatments for a particular patient, then $Y_{t+1}[\bar{a}_t] = Y_{t+1}$. This implies that the potential outcome under the treatment sequence $\bar{a}_t$ corresponds to the observed (factual) outcome for the patient, conditional on $\bar{A}_t = \bar{a}_t$.

Mapping to offline RL, consistency means that for any given action $a_t$ the observed next state $s_{t+1}$ and returns-to-go $g_{t+1}$ reflect the true outcome of the action. In the context of offline RL this assumption holds given that observational data is collected from behaviour policies $\pi_\beta$ that were trained in the same environment; therefore, the data reflects the actual dynamics of the environment.

**Assumption A.2 (Sequential Overlap)**: For every history, there is always a non-zero probability of receiving or not receiving any treatment over time:

$$0 < P(A_t = a_t | \bar{H}_t = \bar{h}_t) < 1, \quad \text{if} \quad P(\bar{H}_t = \bar{h}_t) > 0,$$

where $\bar{h}_t$ is a particular historical experience.

For offline RL, sequential overlap guarantees that for any observed history $h_t$, every action $a_t$ has a non-zero probability of being chosen. This assumption is met if $D_{\text{env}}$ provides adequate coverage of the state-action space. If the behaviour policy $\pi_\beta$ used to collect the data explores a wide range of actions under different histories, we can reasonably assume that the sequential overlap condition hold.

**Assumption A.3 (Sequential Ignorability)**: This states that the current treatment is independent of the potential outcome, given the observed history:

$$A_t \perp Y_{t+1}[a_t] \mid \bar{H}_t, \forall a_t.$$

This means there are no unmeasured confounders that simultaneously influence both the treatment and the outcome.

Sequential ignorability implies that the observed history $h_t$ includes all relevant information that influences both the agent's actions and the potential future outcomes. Since we only perform counterfactual reasoning on observed data in $D_{\text{env}}$, we rely on the assumption that the dataset sufficiently captures the relevant factors affecting the treatments and the resulting outcomes.

In prior works, these assumptions are applied to both environments with discrete or continuous treatments (Melnychuk et al., 2022; Frauen et al., 2023; Bahadori et al., 2022)

## C  DETAILS OF DT LEARNING TO REASON COUNTERFACTUALLY

---

**Algorithm 1** Learrning to Reason Counterfactually Algorithm

---

**Require:** Offline environment dataset $D_{\text{env}}$.
1: **Initialize:** Treatment model $\mathcal{T}$, Outcome model $\mathcal{O}$.
2: **for** $k = 1, \ldots, K$ **do**
3:     Sample batch: $\tau = \left(g_t^{(i)}, s_t^{(i)}, a_t^{(i)}\right)_{t=1}^{T}, \quad i = 1, 2, \ldots, N$ from $D_{\text{env}}$.
4:     Update $\mathcal{T}$ by minimizing loss $\mathcal{L}_{\mathcal{T}(\theta)}$ with Eq. 4 or Eq. 5 using data from $\tau$.
5:     Update $\mathcal{O}$ by minimizing loss $\mathcal{L}_{\mathcal{O}(\eta)}$ with Eq. 7 using data from $\tau$.
6: **end for**

---

## D  DETAILS OF DT COUNTERFACTUAL REASONING

---

**Algorithm 2** DT Counterfactual Reasoning Algorithm

---

**Require:** Offline environment dataset $D_{\text{env}}$, Treatment model $\mathcal{T}$, Outcome model $\mathcal{O}$, number of action sampled $n_a$, number of experiences wanted $n_e$, and function $U^{\alpha}(\mathbf{S}_{t+1})$ from Eq. 10.
1: **Initialize:** Counterfactual experience buffer $D_{\text{crdt}}$.
2: **for** $k' = 1, \ldots, K'$ **do**
3:     Sample batch: $\tau' = \left(g_t^{(i)}, s_t^{(i)}, a_t^{(i)}\right)_{t=1}^{T}, \quad i = 1, 2, \ldots, N'$ from $D_{\text{env}}$.
4:     **for** $\tau'^{(i)}$ in $\tau$ **do**
5:         **for** $t = \frac{T}{2}$ to $T$ **do**
6:             Init: $h_{t-1} = (g_1, s_1, a_1, \ldots, g_{t-1}, s_{t-1}, a_{t-1})$.
7:             $\hat{a}_t^{(j)} \leftarrow \mathcal{T}(h_{t-1}, s_t, g_t; \theta), \quad j = 1, 2, \ldots, n$ ▷ Sample $n_a$ counterfactual treatments.
8:             Init: $\hat{h}_t^{(j)} = (g_1, s_1, a_1, \ldots, g_t, s_t, \hat{a}_t^{(j)})$.
9:             $\hat{s}_{t+1}^{(j)}, \hat{g}_{t+1}^{(j)} \leftarrow \mathcal{O}(\hat{h}_t^{(j)})$.
10:             **if** $\hat{g}_{t+1} < g_{t+1}$ and not $U^{\alpha}(\mathbf{S}_{t+1})$ **then**               ▷ Check for all $\hat{a}_t^{(j)}$.
11:                 $a_t, s_{t+1}, g_{t+1} = \hat{a}_t, \hat{s}_{t+1}, \hat{g}_{t+1}$.
12:             **else**                                                                ▷ If all $\hat{a}_t^{(j)}$ fail.
13:                 Break.
14:             **end if**
15:             **if** $t = T$ and $\text{len}(D_{\text{crdt}}) < n_e$ **then** $D_{\text{crdt}} \leftarrow \tau'^{(i)}$.
16:             **end if**
17:         **end for**
18:     **end for**
19: **end for**

---

### D.1  COUNTERFACTUAL ACTION SELECTION IN CONTINUOUS ACTION SPACE

We aim to select $n_a$ actions as our counterfactual actions. The selection of these actions in a continuous action space environment is inspired by the theory of maximum Gaussian random variables (Kamath, 2015). The expectation of maximum of Gaussian random variables are bounded as:

$$0.23\sigma \cdot \sqrt{\ln(n)} \leq \mathbb{E}\left[\max(x - \mu)\right] \leq \sqrt{2}\sigma \cdot \sqrt{\ln(n)}.$$

where $\mu$ is the mean of the distribution and $\sigma$ is the standard deviation. Applying this equation to our approach, wherein continuous action is assumed to follow a normal Gaussian distribution. Thus, for an action $a_t$, at timestep $t$, we can rewritten the equation into:

$$0.23\sigma \cdot \sqrt{\ln(n)} \leq \mathbb{E}\left[\max(a_t - \mu_t)\right] \leq \sqrt{2}\sigma_t \cdot \sqrt{\ln(n)},$$

or

$$0.23\sigma \cdot \sqrt{\ln(n)} + \mu_t \leq \mathbb{E}\left[\max(a_t)\right] \leq \sqrt{2}\sigma_t \cdot \sqrt{\ln(n)} + \mu_t.$$

We choose to use the upper bound of this equation as the bound for our outlier actions, thus, from this bound, we will start searching for a number of $n_a$ outlier actions. The bound can be written as:

$$E\left[\max(a_t)\right] \leq \mu_t + \sqrt{2}\sigma_t \sqrt{\ln(n_{enc})}.$$

As $\Phi^{-1}(0.08) \approx -\sqrt{2}$. We can derive our formula to calculate each action:

$$a_t = \mu_t - \Phi^{-1}(0.08 - j \cdot \beta)\sigma_t\sqrt{\ln(n_{enc})}.$$

where $\beta$ is the step size and $j = 0, 1, \ldots, n_a$ indicates the index of the $j$-th action from the total $n_a$ sampled counterfactual actions. $\Phi^{-1}$ is the quantile function of the standard normal distribution. When $j = 0$, the value of $\Phi^{-1}(0.08 - j \cdot \beta) = \Phi^{-1}(0.08) \approx -\sqrt{2}$, thus:

$$\mu_t - \Phi^{-1}(0.08)\sigma_t\sqrt{\ln(n_{enc})} \approx \mu_t + \sqrt{2}\sigma_t\sqrt{\ln(n_{enc})}.$$

Here, $n_{enc}$ denotes the number of times the model has encountered an input $(h_t, s_{t+1}, g_{t+1})$. In a continuous environment, recording the counting for such input is difficult. Thus, we employed a hashing function, specifically, we used the hashlib.md5() hashing function [3] in our implementation to record the input as key and the counting as the value in a dictionary. As MD5 hashing looks for an exact match of data, we expect that such hashing process will only help with saving memory and not affect the overall result of the method.

## D.2    COMPUTE THE MAXIMUM VARIANCE BETWEEN PREDICTIONS

In this section, we present our method that was used to compute the maximum variance of the predictions in $\mathbf{S}_{t+1}$ using the function $\mathrm{Var}(\mathbf{S}_k)$. $\mathbf{S}_{t+1} = \begin{bmatrix} s_{t+1}^{(1)} & s_{t+1}^{(2)} & \cdots & s_{t+1}^{(m)} \end{bmatrix}$ is the matrix of state predictions at timestep $t+1$, output from $m$ forward passes of the Outcome model $\mathcal{O}$ that was trained with dropout regularization layers.

Thus, $\mathbf{S}_{t+1} \in \mathbb{R}^{m \times d}$, where $m$ is the number of predictions and $d$ is the dimension of each prediction. Each row of $\mathbf{S}_{t+1}$, denoted as $\mathbf{s}_{t+1}^{(i)} = \begin{bmatrix} s_{t+1}^{(i,1)} & s_{t+1}^{(i,2)} & \cdots & s_{t+1}^{(i,d)} \end{bmatrix}$, represents a predicted state at timestep $t+1$, where $i = 1, 2, \ldots, m$. Each $\mathbf{s}_{t+1}^{(i)}$ is a $d$-dimensional vector representing the state in the predicted space.

The variance for each dimension of the predicted states is computed using the covariance matrix of $\mathbf{S}_{t+1}$. The covariance matrix $\Sigma_k \in \mathbb{R}^{d \times d}$ is defined as:

$$\Sigma_k = \frac{1}{m-1}\sum_{i=1}^{m}\left(\mathbf{s}_{t+1}^{(i)} - \bar{\mathbf{s}}_{t+1}\right)\left(\mathbf{s}_{t+1}^{(i)} - \bar{\mathbf{s}}_{t+1}\right)^T,$$

where $\bar{\mathbf{s}}_{t+1}$ is the mean of the predicted states, $\bar{\mathbf{s}}_{t+1} = \frac{1}{m}\sum_{i=1}^{m}\mathbf{s}_{t+1}^{(i)}$.

The variance for each dimension $j'$ (for $j' = 1, 2, \ldots, d$) is then extracted from the diagonal elements of $\Sigma_k$, denoted as:

$$\mathrm{Var}(\mathbf{s}_{t+1}^{(j')}) = \Sigma_{k,j'j'}.$$

This allows us to get the maximum variance across all dimensions:

$$\mathrm{Var}(\mathbf{S}_{k=t+1}) = \max\left(\Sigma_{k,11}, \Sigma_{k,22}, \ldots, \Sigma_{k,dd}\right).$$

---

[3]https://docs.python.org/3/library/hashlib.html

In environments with image observation space such as Atari games, calculating the covariance matrix from raw observations is computationally expensive. Thus, we use the encoded observations, from the Outcome model, to form the prediction matrix instead. Thus $\mathbf{S}_{t+1} = \begin{bmatrix} \phi(s)_{t+1}^{(1)} & \phi(s)_{t+1}^{(2)} & \cdots & \phi(s)_{t+1}^{(m)} \end{bmatrix}$, where $\phi(s)$ denotes the encoding.

### D.3 CHOOSING THE UNCERTAINTY THRESHOLD

Our strategy to determine the uncertainty threshold $\alpha$ for each testing environment and dataset is inspired by the process used in (Kidambi et al., 2020). Specifically, we compute the accumulated maximum variance $\sum_{k=t_0}^{t+1} \max\left(\text{Var}(\mathbf{S}_k)\right)$ over several batch data (we use 1000 samples in this paper) sampled from the static dataset $D_{env}$. Then, we compute the mean $\mu_d$, the standard deviation $\sigma_d$ over all the accumulated maximum variance that we have collected. The uncertainty threshold is then $\alpha = \mu_d + \sigma_d \cdot \varsigma$. We tune the value of $\varsigma$ in steps of 0.5. The final uncertainty threshold $\alpha$ for each environment is presented in Appendix. F.11.

## E DETAILS OF DT OPTIMIZE DECISION-MAKING WITH COUNTERFACTUAL EXPERIENCE

---

**Algorithm 3** Optimize Decision-Making with Counterfactual Data Algorithm

---

**Require:** Offline environment dataset $D_{\text{env}}$, Counterfactual experience buffer $D_{\text{crdt}}$.

1: **Initialize:** $\mathcal{M}$ agent.
2: **for** $k = 1, \ldots, K$ **do**
3:      Sample batch: $\tau = \left(g_t^{(i)}, s_t^{(i)}, a_t^{(i)}\right)_{t=1}^{T}, \quad i = 1, 2, \ldots, N$ from $D_{\text{env}}$.
4:      Calculate loss $\mathcal{L}_{\mathcal{M}(\delta)}^{\text{env}}$ with Eq. 11 or Eq. 12 using data from $\tau$.
5:      Sample batch: $\tau' = \left(g_t^{(i)}, s_t^{(i)}, a_t^{(i)}\right)_{t=1}^{T}, \quad i = 1, 2, \ldots, N'$ from $D_{\text{crdt}}$.
6:      Calculate loss $\mathcal{L}_{\mathcal{M}(\delta)}^{\text{crdt}}$ with Eq. 11 or Eq. 12 using data from $\tau'$.
7:      Update $\mathcal{M}$ by minimizing loss $\mathcal{L}_{\mathcal{M}(\delta)} = \mathcal{L}_{\mathcal{M}(\delta)}^{\text{env}} + \mathcal{L}_{\mathcal{M}(\delta)}^{\text{crdt}}$.
8: **end for**

---

## F ADDITIONAL EXPERIMENT DETAILS

### F.1 DETAILS OF BASELINES

In our paper, we have compare CRDT against a number of baselines including including conventional RL and sequential modeling techniques. Conventional methods include Behavior Cloning (BC) (Pomerleau, 1988), model-free offline methods, such as Conservative Q-Learning (CQL) (Kumar et al., 2020) and Implicit Q-Learning, (IQL) (Kostrikov et al., 2021b) and model-based offline methods, such as MOPO (Yu et al., 2020) and MOReL (Kidambi et al., 2020). Sequential modeling baselines include simple backbone DT (Chen et al., 2021), Elastic Decision Transformer (EDT) (Wu et al., 2024) and state-of-the-art Reinformer (REINF) (Zhuang et al., 2024). In this section, we will clarify which results we have get from the original paper, and which results we have reproduced and where the source code is from. Given limited computational resources, our focus is on reproducing the result of sequential modeling approaches, which are our direct comparing baselines.

- The results of BC in Table. 1 comes from the REINF paper (Zhuang et al., 2024), whereas the results of BC in Table. 2 is from the original DT paper (Chen et al., 2021).

- The results of model-free offline RL methods, CQL and IQL, are in Table. 1, also comes from the REINF paper (Zhuang et al., 2024). While the results of model-based offline RL methods, MOPO and MOReL, are obtained straight from their original papers (Yu et al., 2020; Kidambi et al., 2020).

- The results of EDT and REINF, in Table. 1, are reproduced using the source codes provided by the authors (MIT licence)[4] for all the Locomotion tasks and Ant tasks, using the hyperparameters that were provided in the associated papers. For REINF, we also ran the source code on the Ant environments for a comprehensive comparison, the hyperparameters that were used are the default hyperparameters that come with the code. For Maze tasks in Fig.2, we reproduce the results of REINF using the hyperparameters provided in the paper.

- All the results of DT are reproduced using the source code provided by the authors (MIT licence)[5].

### F.2 DETAILS OF DATASET AND ENVIRONMENTS

We compare our CRDT algorithm against baselines on several datasets. These include those with continuous action space environments and those that come with discrete action space environments. This is to provide a comprehensive test for the Counterfactual Action Selection and the Counterfactual Action Filtering mechanism. In this section, we provide an overview of the testing environment.

Continuous action space environments include Locomotion, Ant, and Maze2d tasks from the D4RL benchmark (Fu et al., 2020). The environments within Locomotion include hopper, halfcheetah and walker. For each of the Locomotion environments and the Ant environments, we have 3 types of dataset medium-replay (med-rep), medium (med), and medium-expert (med-exp). The environments within Maze2d environments include large and umaze; each of these environments has its corresponding dataset maze2d-large and maze2d-umaze. We obtain the datasets for Locomotion and Maze tasks using the code associated with the Reinformer paper (Zhuang et al., 2024), while, the dataset for Ant tasks are collected using the code associated with the Elastic Decision Transformer paper (Wu et al., 2024). We evaluate our algorithms using gym environments from gym package ver. 0.18.3 (Brockman, 2016).

Discrete action space environments include Breakout, Qbert, Pong and Seaquest. The data for these environments were collected using the code provided in the original Decision Transformer paper (Chen et al., 2021). We also use the evaluation code provided in this paper to evaluate our algorithm. Specifically, we use ale-py package ver. 0.8.1 (Bellemare et al., 2013) for evaluation.

### F.3 HOW IS THE PERFORMANCE OF CRDT ON MODIFIED EVALUATING ENVIRONMENTS?

We further evaluate CRDT's generalizability by testing its performance in modified environments, where the dynamics differ from those in the $D_{env}$ dataset generated by the behaviour policy $\pi_\beta$. Out

---

[4]https://github.com/kristery/Elastic-DT, https://github.com/Dragon-Zhuang/Reinformer

[5]https://github.com/kzl/decision-transformer

Table 4: Performance comparison on modified evaluating environments. We report the results over 5 seeds. For each seed, evaluation is conducted over 100 episodes. The best result is shown in **bold**.

| Dataset | Modification | DT | REINF | CRDT (Ours) |
|---------|--------------|-----|-------|-------------|
| hopper-med-rep | head | 326.5 | 348.0 | **359.54 ± 47.5** |
| hopper-med-rep | thigh | **2930.5** | 2841.6 | 2879.4 ± 421.5 |
| halfcheetah-med-rep | head | 582.1 | 371.6 | **617.2 ± 32.3** |
| halfcheetah-med-rep | thigh | 1966.6 | 1345.2 | **2070.8 ± 264.6** |

of the four environments tested, in Table. 4, CRDT improves DT's performance in three. In contrast, REINF shows weaker results in these environments, likely due to its architecture, which forces it to maximize returns within $D_{\text{env}}$—a condition that may not hold in the modified environments. CRDT excels compared to the original DT method because it generates additional counterfactual experiences, enabling it to cover a broader range of scenarios than the $D_{\text{env}}$ dataset alone.

## F.4 ATARI RAW SCORES

Table 5: Performance comparison (raw score) on various Atari games. We report the results over 3 seeds. For each seed, evaluation is conducted over 10 episodes. The best result is shown in **bold**. ♠ indicates games in which CRDT improves the backbone DT approach.

| Game | BC | DT | CRDT (Ours) |
|------|-----|-----|-------------|
| Breakout | 138.9 ± 17.3 | 57.6 ± 1.5 | **71.7 ± 18.5**♠ |
| Qbert | **2464.1 ± 1948.2** | 1118.6 ± 195.6 | 1155.3 ± 89.2♠ |
| Pong | 9.7 ± 7.2 | **29.5 ± 1.9** | 15.8 ± 3.34 |
| Seaquest | 968.6 ± 133.8 | 2494.0 ± 2732.6 | **3190.6 ± 264.6**♠ |

Table 6: Atari Baseline Scores.

| Game | Random | Gamer |
|------|--------|-------|
| Breakout | 2 | 30 |
| Qbert | 164 | 13455 |
| Pong | -21 | 15 |
| Seaquest | 68 | 42055 |

We present the raw score of the experiments on Atari games in Table. 5. These results correspond to the normalized results presented in Table. 2. For the purpose of normalization, we used the data in Table. 6. This is similar to the process of normalization that have been used in Chen et al. (2021).

## F.5 CHANGING COUNTERFACTUAL EXPERIENCE SIZE

We conduct this experiment to show the impact of varying the number of counterfactual experiences $n_e$ recorded in $D_{\text{crdt}}$. The experiment was conducted on the 10% of the walker2d-medium-replay dataset. Our expectation is that a higher number of experiences the higher the performance. We evaluate the performance with 4000 samples (corresponding to the 10% result in Fig. 2(c)), 8000 samples, and 16000 samples; the result is presented in Fig. F.1. The figure reveals an upward trend in performance as the number of recorded samples increases, validating our expectation. With 16000 samples, CRDT achieves approximately 59 points (10 points higher than when using 4000 samples), closely approaching the performance of DT trained on the entire dataset (approximately 62 points as shown in Table. 1). However, the performance gains also diminish as the number of samples increases. While the improvement from 4000 to 8000 samples is around 6 points, the increase from 8000 to 16000 samples is only about 3 points. Moreover, generating more counterfactual experiences demands greater computational resources, underscoring the balance between performances and computational resources.

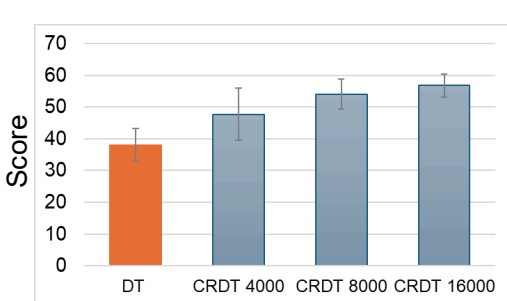

Figure F.1: The impact of varying the number of counterfactual experiences $n_e$ in $D_{\mathrm{CRDT}}$ on the performance. The agent is trained using the 10% walker2d-medium-replay dataset. The terms CRDT 4000, 8000, and 16000 refer to configurations of CRDT with $n_e$ set to 4000, 8000, and 16000 samples, respectively. We report the results over 5 seeds. For each seed, evaluation is conducted over 100 episodes.

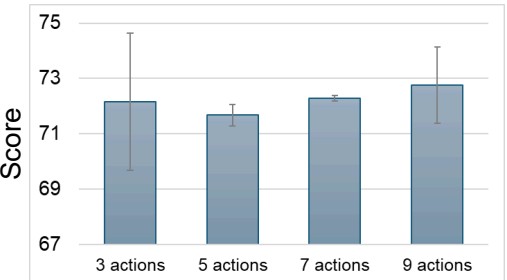

Figure F.2: The impact of varying the number of search action $n_a$. The agent is trained with walker2d-medium-replay dataset. We report the results over 5 seeds. For each seed, evaluation is conducted over 100 episodes.

### F.6 CHANGING NUMBER OF SEARCH ACTIONS

We conduct an additional experiment to assess the impact of varying the number of search actions, $n_a$, on the walker2d-medium-replay dataset. We specifically test 3, 5, 7, and 9 actions, with the results presented in Fig. F.2. As shown, increasing the number of actions generally improves the

performance of the DT agent, aside from an outlier when $n_a = 3$. However, using $n_a = 3$ also results in a significantly higher variance in performance and produces the lowest score among the four configurations. This finding aligns with our expectation that increasing the number of actions would broaden the diversity of covered states, enabling the agent to learn more about the environment and improve its performance.

### F.7 TOY ENVIRONMENT RESULTS

Table 7: Performance comparison on the toy environment in Fig. 1. The dataset ratio is between the number of green trajectories versus the number of blue trajectories.

| Dataset | DT | CRDT (Ours) |
|---------|-----|-------------|
| 10:1 | 0.37±0.30 | **0.83±0.14** |
| 20:1 | 0.41±0.36 | **0.90±0.07** |
| 50:1 | 0.39±0.18 | **0.92±0.15** |

The result is provided in Table. 7, corresponding to the analysis in Sect. 4.4.2.

### F.8 CRDT (REINF) AND CRDT (EDT)

In Table 8, we present the results of using CRDT with REINF (Zhuang et al., 2024) and EDT (Wu et al., 2024) as the backbone algorithms. A note here is that we only replace decision-making agent $\mathcal{M}$ with the new backbone and not model $\mathcal{T}$ and $\mathcal{O}$.

**CRDT (REINF)**

Although CRDT with REINF shows slight improvements over CRDT with the original DT on the Locomotion and Ant tasks, its performance is significantly lower on the Maze2d tasks. We attribute this decline in performance to the increased difficulty of the Maze2d tasks. Additionally, the underlying REINF algorithm likely requires parameter tuning, especially when integrating new counterfactual experiences. This tuning was not conducted in our study, which may have led to the observed decrease in performance. Here, we used the original parameters provided in the REINF paper for the backbone algorithm. Overall, CRDT with the Reinformer backbone still improve the results of REINIF, as presented in Table 1, albeit only marginally.

**CRDT (EDT)**

Similarly, the result of using EDT as the decision-making also indicates an improve in performance over Locomotion tasks when comparing to the EDT's results provided in Table 1. We saw a noticeable improvement on walker2d-med-rep task of approximately 20%. The result on Ant tasks indicates a marginally improvement. The result overall performance, however, is still not as good as when using CRDT (DT) or CRDT (REINF).

### F.9 COMPARISON ON RANDOM DATASET

Refer to Table. 9, we compare the performance of CRDT against other sequential modelling methods on the random dataset. CRDT outperforms other methods on halfcheetah and walker2d environments. A note here is that we did not perform parameters tuning for REINF and EDT, but used the suggested parameters for med-rep dataset from their papers. Interestingly, DT performs unexpectedly well on hopper-rand, which is a noteworthy observation.

### F.10 VISUALIZING THE DISTRIBUTION OF COUNTERFACTUAL ACTIONS AND ORIGINAL ACTIONS

We refer to Fig. F.3 and F.4, where we illustrate the frequency distribution of action values across dimensions in the walker2d-med-rep and halfcheetah-med-exp respectively, between the counterfactual and the original actions. We compute the value over the whole original dataset provided by D4RL, while for the counterfactual samples, we compute the value over 4000 samples. One can see that in Fig. F.3, the distribution across the last 5 dimensions differs, while in Fig. F.4, the differences are in all 6 dimensions. We hypothesize that these significant distribution differences

Table 8: Performance comparison between CRDT (DT) versus CRDT (REINF) versus CRDT (EDT) on Locomotion, Ant, and Maze tasks. We report the results over 5 seeds. For each seed, evaluation is conducted over 100 episodes.

| Dataset | Sequence Modeling Methods | | |
|---|---|---|---|
| | CRDT (DT) | CRDT (REINF) | CRDT (EDT) |
| halfcheetah-med | 42.8±2.32 | 43.0±1.51 | 43.1±0.36 |
| halfcheetah-med-rep | 38.0±2.54 | 36.8±2.01 | 36.0±2.21 |
| halfcheetah-med-exp | 96.4±2.32 | 94.4±1.74 | 72.3±9.19 |
| hopper-med | 67.9±1.56 | 74.2±6.37 | 54.4±7.56 |
| hopper-med-rep | 85.5±3.24 | 85.2±2.29 | 70.2±8.71 |
| hopper-med-exp | 110.3±0.14 | 110.3±0.63 | 108.7±2.92 |
| walker2d-med | 78.9±0.91 | 79.2±2.73 | 65.4±1.51 |
| walker2d-med-rep | 72.2±0.11 | 70.0±2.29 | 72.6±21.7 |
| walker2d-med-exp | 109.05±0.63 | 108.7±0.46 | 107.2±0.22 |
| **Total Locomotion** | 701.38±1.53 | 701.88±2.22 | 630.2±6.29 |
| ant-med-rep | 91.0±8.84 | 92.1±0.55 | 87.2±3.57 |
| ant-med | 95.84±8.32 | 95.2±1.13 | 90.2±4.60 |
| **Total Ant** | 186.8±8.58 | 187.3±0.84 | 177.4±4.08 |
| maze2d-umaze | 55.2±9.20 | 41.3±4.39 | - |
| maze2d-large | 42.3±3.74 | 47.7±13.6 | - |
| **Total Maze2d** | 97.5 ±6.47 | 89 ±8.99 | - |

Table 9: Performance comparison between DT, REINF, EDT, CRDT on random D4RL dataset. We report the results over 3 seeds. For each seed, evaluation is conducted over 100 episodes.

| Dataset | Sequence Modeling Methods | | | |
|---|---|---|---|---|
| | DT | EDT | REINF | CRDT |
| halfcheetah-rand | 2.01±2.27 | 0.82±2.58 | - | **2.21±2.28** |
| hopper-rand | 10.5±0.27 | 3.97±0.39 | 9.98±0.30 | 9.59±0.44 |
| walker2d-rand | 1.20±0.10 | 0.77±0.35 | 0.71±0.17 | **2.60±0.42** |

may have contributed to the greater improvement in the walker2d-med-rep and halfcheetah-med-exp environments, as demonstrated in Table 1.

### F.11 DETAILS OF HYPERPARAMETERS

In this paper, we have introduced a number of new parameters. This is divided into those that were used in discrete action space environments and those that were used in continuous action space environments. Apart from these parameters, we also have the parameters of the backbone DT algorithm. The same hyperparameters were used for the Treatment model $\mathcal{T}$, the Outcome model $\mathcal{O}$ and the agent $\mathcal{M}$.

**Continuous Action Space Environments**

We follow the hyperparameters proposed in the original paper by Chen et al. (2021), apart from those being specified. These parameters are applied to all of the 3 models and are provided in Table. 10.

Table 10: DT's Parameters for Continuous Action Space Environments.

| **Dataset** | Batch Size | $K$ | Learning Rate | No. Layers | Atten. Heads |
|---|---|---|---|---|---|
| maze2d-large | 64 | 10 | 0.0004 | 5 | 8 |
| maze2d-umaze | 64 | 20 | 0.0001 | 3 | 8 |
| Others | 64 | 20 | 0.0001 | 3 | 1 |

Additional parameters that we have introduced in this paper include the number of search actions $n_a$, the step size $\beta$ when searching for the action, the uncertainty threshold $\alpha$, and the number of experiences $n_e$. For simplicity, we opt for using a step size $\beta = 0.01$ for all environments. The

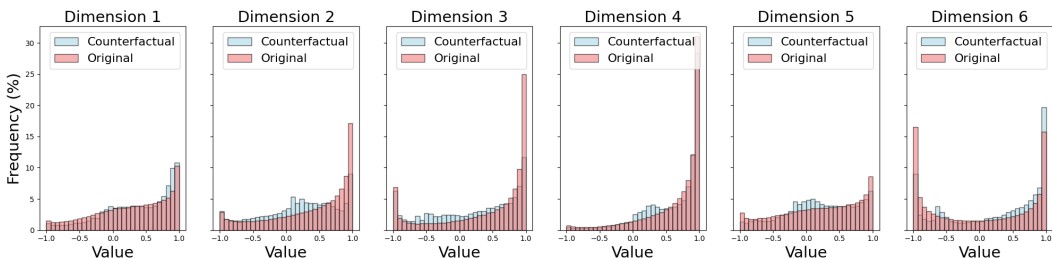

Figure F.3: Frequency distribution of action values across dimensions in the walker2d-med-rep environment. The histograms represent the percentage frequency of action values for each of the six dimensions, offering insights into the distribution patterns of actions in the dataset.

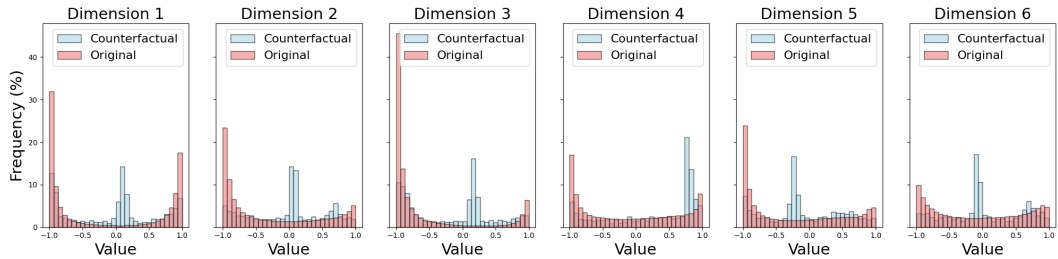

Figure F.4: Frequency distribution of action values across dimensions in the halfcheetah-med-exp environment. The histograms represent the percentage frequency of action values for each of the six dimensions, offering insights into the distribution patterns of actions in the dataset.

parameter $\alpha$ is determined through the process outlined in Appendix D.3. These parameters are provided in Table. 11.

Table 11: New Hyperparameters for Continuous Action Space Environments.

| Dataset | $n_a$ | $\alpha$ | $n_e$ |
|---|---|---|---|
| halfcheetah-med-rep | 5 | 4.2 | 1000 |
| halfcheetah-med | 7 | 2.5 | 1000 |
| halfcheetah-med-exp | 5 | 0.3 | 1000 |
| hopper-med-rep | 5 | 0.7 | 1000 |
| hopper-med | 7 | 0.7 | 4000 |
| hopper-med-exp | 5 | 0.4 | 4000 |
| walker2d-med-rep | 7 | 1.8 | 4000 |
| walker2d-med | 5 | 1.8 | 1000 |
| walker2d-med-exp | 5 | 0.4 | 4000 |
| ant-med-rep | 5 | 0.8 | 4000 |
| ant-med | 5 | 1.5 | 2000 |
| maze2d-umaze | 5 | 0.1 | 2000 |
| maze2d-large | 5 | 0.1 | 2000 |
| halfcheetah-med-rep (less_data) | 5 | 0.1 | 4000 |
| hopper-med-rep (less_data) | 5 | 0.7 | 4000 |
| walker2d-med-rep (less_data) | 7 | 1.8 | 4000 |
| maze2d-umaze (less_data) | 5 | 0.1 | 4000 |
| maze2d-large (less_data) | 5 | 0.1 | 4000 |

**Discrete Action Space Environments**

For discrete action space environments (Atari), we follow the hyperparameters proposed in the original paper by Chen et al. (2021) and apply it to all 3 models. The selected parameters are provided in Table 12.

Table 12: DT's Parameters for Discrete Action Space Environments.

| **Games** | $K$ | Learning Rate | No. Layers | Atten. Heads |
|---|---|---|---|---|
| Breakout, Qbert, Seaquest | 30 | 0.0006 | 6 | 8 |
| Pong | 50 | 0.0006 | 6 | 8 |

Table 13: New Hyperparameters for Discrete Action Space Environments.

| **Games** | $\alpha$ | $n_e$ |
|---|---|---|
| Pong, Seaquest, Breakout | 10 | 500 |
| Qbert | 75 | 500 |

We introduce three key parameters: the outlier action threshold $\gamma$, the number of experiences $n_e$, and the uncertainty threshold $\alpha$. As in continuous action space environments, the uncertainty threshold $\alpha$ is determined using the method described in Appendix D.3. The action threshold $\gamma$ is tuned over the range $[0.1, 0.3]$ with a step size of $0.05$, and a value of $0.25$ is selected for all four evaluation environments. For $n_e$, a value of 500 transitions is chosen, constrained by available computational resources. The selected parameters are summarized in Table 13.

## G    RELATION TO CAUSAL INFERENCE AND COUNTERFACTUAL REASONING

Although CRDT is inspired by causal inference and counterfactual reasoning, the method did not explicitly establish a formal causal structure learning process, such as constructing a causal graph or a Structural Causal Model (SCM) (Pearl & Mackenzie, 2018). The method is more closely related to the potential outcome framework (Neyman, 1923; Rubin, 1978) and its extension to time-varying treatments and outcomes (Robins & Hernan, 2008), which did not explicitly require a causal graph (Pearl & Mackenzie, 2018). The proposed counterfactual reasoning process in CRDT also differs from "Pearl-style counterfactual reasoning", which requires the inference of the posterior distribution of exogenous noise variable and intervention on the parental variables. In CRDT, we assume that the noise is implicit in the dynamic model. The method, however, leverages several concepts from these frameworks.

Specifically, our method estimates the outcomes of different treatments using an Outcome Network, which aligns with prior work in adapting machine learning methods for causal effect inference (Shalit et al., 2017; Jacob, 2021; Melnychuk et al., 2022), where neural networks were used to estimate treatment effects by modeling counterfactual outcomes. While the potential outcome framework does not strictly require a causal graph and the choice of the underlying ML algorithm is very flexible (Jacob, 2021), in CRDT, we purposefully chose Transformers architecture for both the Treatment and Outcome networks due to their ability to capture long-term dependencies through attention mechanisms. The attention scores within the Transformer architecture underpinning these networks can serve as a simple causal masking mechanism (Pitis et al., 2020; Seitzer et al., 2021; Pitis et al., 2022). Furthermore, our framework assumes the three key causal assumptions, namely Consistency, Sequential Overlap, and Sequential Ignorability, as detailed in Appendix B. These connections demonstrate how causal inference concepts underpin our framework, even if they are not formalized in the traditional sense.

