# OpenReview forum: "Beyond the Known: Decision Making with Counterfactual Reasoning Decision Transformer"
_ICLR.cc/2025/Conference — Submitted to ICLR 2025_

### Official Review · Reviewer_cDB6 · 2024-11-03

**Soundness:** 3
**Presentation:** 3
**Contribution:** 2
**Rating:** 3
**Confidence:** 3

**Summary:**

In this paper, a novel framework called the Counterfactual Reasoning Decision Transformer (CRDT) is proposed. Inspired by counterfactual reasoning, CRDT elevates DT's capacity to extrapolate beyond familiar data by creating and leveraging counterfactual experiences. This enhancement enables superior decision-making in unfamiliar scenarios. Experiments conducted on Atari and D4RL benchmarks, encompassing scenarios with limited data and modified dynamics, illustrate that CRDT surpasses traditional DT methods. Furthermore, engaging in counterfactual reasoning empowers the DT agent with stitching abilities, enabling the fusion of suboptimal trajectories without structural alterations.

**Strengths:**

The problem raised in this paper is important and the experiments are solid.

**Weaknesses:**

1) The basic idea in this paper is to encourage the agent to explore the counterfactual actions according to the training data. The authors realized this by a pretrained dynamics model (outcome model). However, the predictions of the dynamics model over these counterfactual actions would introduce large biases because these actions would not have enough data to support. Then the proposed method seems to be unreasonable.

2) The relationship between the proposed method in this paper and those model-based offline RL algorithms. The core problem is to solve the problem of learning from sub-optimal data by the stitching abilities (e.g. the example in Fig. 1), but, as far as we have known, the model-based offline RL algorithms such as MOPO [1] and MOBILE [2] may also have the abilities of stitching. We wish the authors to discuss the performance of model-based offline RL algorithms in the example of Fig. 1.

[1] MOPO: Model-based Offline Policy Optimization
[2] Model-Bellman Inconsistency for Model-based Offline Reinforcement Learning

3) Lack of the discussion about the advantages over the currently stitching-enabled methods, such as [3], [4] and [5]. Why would they fail the task as is listed in Fig. 1?

[3] Free from Bellman Completeness: Trajectory Stitching via Model-based Return-conditioned Supervised Learning
[4] Elastic decision transformer
[5] Reinformer: Max-return sequence modeling for offline rl


4) A shortcoming of the proposed Counterfactual Action Filtering. If the true dynamics model is stochastic, then even the actions are safe to be taken, the sum of the variance of the trajectory would also be high and the U in Eq.(10) would also be large. In such case, such actions would be filtered, but in fact they may be safe to be taken.

5) This paper aims to research the topic of offline rl from sub-optimal dataset, then the 'random' datasets of D4RL may be important. So the authors are encouraged  to give some experimental results on 'random' datasets in Table 1.

6) Concerning the counterfactual action selection, how the selected action is related to the performance improvement should be further formulated and explained.

**Questions:**

see above

---

> ### Author Response · Authors · 2024-11-21
> **Response to Reviewer cDB6 (1/2)**
>
> Dear Reviewer cDB6,
>
> We sincerely appreciate your constructive feedback and the time you’ve dedicated to reviewing our work. We are grateful for your recognition of the importance of the problem we tried to address and our efforts in conducting extensive experiments. We hope the following responses will clarify your remaining concerns. However, we are open to further discussion if you have any questions.
>
>
>
>
> - **W1** (On Addressing Biases when Exploring Low Support Regions): We would like to emphasize that we have carefully addressed this issue by introducing **Counterfactual Action Filtering, which filters out treatments where the Outcome network exhibits high uncertainty**. Our method aims to maximize the utility of the given dataset, even when it may be suboptimal, by selecting actions that are less likely to be chosen but potentially beneficial (resulting in higher returns). Most importantly, these actions are only considered if the model is confident in its predictions (i.e., when uncertainty is low).
>
> - **W2** (On Stitching Ability): We appreciate the reviewer’s focus on the stitching ability of DT, as highlighted in both Weaknesses 2 and 3, recognizing this as a limitation of DT. However, we would like to clarify that \textbf{our primary objective} was to **address data sparsity**. The **stitching ability is an additional benefit** that emerged. The stitching ability of our method was particularly evident when tested in environments, such as the Maze2D environment, which is known to require stitching abilities (as noted in [1]). Model-based RL methods, such as MOPO [2] and MOBILE [3], have stitching ability. However, it is because they use traditional RL algorithms, such as SAC, as their backbones, thus, their stitching ability comes from the TD-learning process of the RL backbones. We, however, improve DT. The generative models method, particularly DT, cannot stitch trajectory because they do not use TD-learning.
>
>
>
> [1] Zhuang Zifeng, Dengyun Peng, Jinxin Liu, Ziqi Zhang, and Donglin Wang. "Reinformer: Max-return sequence modeling for offline rl." arXiv preprint arXiv:2405.08740 2024.
>
> [2] Yu Tianhe, Garrett Thomas, Lantao Yu, Stefano Ermon, James Y. Zou, Sergey Levine, Chelsea Finn, and Tengyu Ma. "Mopo: Model-based offline policy optimization." Advances in Neural Information Processing Systems 33 (2020): 14129-14142.
>
> [3] Sun Yihao, Jiaji Zhang, Chengxing Jia, Haoxin Lin, Junyin Ye, and Yang Yu. "Model-Bellman inconsistency for model-based offline reinforcement learning." In International Conference on Machine Learning, pp. 33177-33194. PMLR, 2023.
>
> - **W3** (On Stitching Ability): While our method may not outperform approaches like [1] or [2] in simpler toy problems (e.g., the example in Figure 1), it demonstrates superior performance in data-sparse stitching-required environments like Maze2D, as shown in Figure 2. Specifically, our method outperforms [1], which we believe represents the current state-of-the-art in stitching for DT. In summary, **the contribution of our approach is to address both the challenges of data sparsity and stitching**, offering a solution that is particularly effective in environments with limited data and stitching requirements. Our approach differs fundamentally from [1] and [2]. While [1] addresses stitching during the training and inference process, our method **solves stitching from the data perspective**. Additionally, our approach stands apart from traditional counterfactual reasoning or model-based offline RL methods, which often prioritize learning a pessimistic dynamic model or select conservative counterfactual action close to the original data. Instead, **our motivation is to select actions that are less likely to occur, but are still beneficial**.
>
> [2] Wu, Yueh-Hua, Xiaolong Wang, and Masashi Hamaya. "Elastic decision transformer." Advances in Neural Information Processing Systems 36 2024.

---

> ### Author Response · Authors · 2024-11-21
> **Response to Reviewer cDB6 (2/2)**
>
> - **W4** (On Stochastic Environments): We agree with the reviewer that our method may not perform well in stochastic environments. However, we believe that addressing stochasticity is beyond the scope of this work. That said, we acknowledge that stochastic environments present a significant challenge, as traditional DT has been shown to underperform compared to traditional offline RL methods in highly stochastic settings, as noted in [4]. This limitation is not unique to our method but is a broader challenge inherent to any supervised learning approach applied to RL environments
>
> [4] Prajjwal Bhargava, Rohan Chitnis, Alborz Geramifard, Shagun Sodhani, and Amy Zhang. When should we prefer decision transformers for offline reinforcement learning? International Conference on Learning Representations, 2024.
>
> - **W5** (On Evaluation on Random Dataset): We have included the evaluation of our method along with other baselines on the random dataset in Table 9 of Appendix F.9. The results are summarized in the table below. **CRDT demonstrates superior performance compared to other methods on halfCheetah and walker2d** (a note is that we did not perform parameters tuning for REINF and EDT, but used the suggested parameters for med-rep dataset from their papers). Interestingly, DT performs unexpectedly well on hopper-rand, which is a noteworthy observation. We are, however, not entirely certain about the cause of DT's superior performance.
> | Dataset            | DT         | EDT        | REINF      | CRDT       |
> |--------------------|------------|------------|------------|------------|
> | halfcheetah-rand   | 2.01±2.27  | 0.82±2.58  | -          | **2.21±2.28** |
> | hopper-rand        | 10.5±0.27  | 3.97±0.39  | 9.98±0.30  | 9.59±0.44  |
> | walker2d-rand      | 1.20±0.10  | 0.77±0.35  | 0.71±0.17  | **2.60±0.42** |
>
> - **W6**: We would like to clarify that the Counterfactual Action Filtering mechanism **selects only actions associated with lower returns-to-go** (line 295), **which correspond to higher returns**. This ensures that the model is trained exclusively on higher-quality data, directly contributing to performance improvement.
>
> We hope the clarification will lead the reviewer to reassess and enhance your score.

---

> > ### Author Response · Authors · 2024-11-27
> > **Following up with Reviewer cDB6**
> >
> > Dear Reviewer cDB6,
> >
> > Thank you again for your valuable feedback. As the author-reviewer discussion period draws to a close, we wanted to ensure that our rebuttal has addressed your concerns. Please let us know if anything requires further clarification or remains unclear.

---

### Official Review · Reviewer_ArKg · 2024-11-05

**Soundness:** 3
**Presentation:** 3
**Contribution:** 2
**Rating:** 5
**Confidence:** 4

**Summary:**

This paper proposes a novel algorithm named Counterfactual Reasoning Decision Transformer to leverage the counterfactual experiences to improve the performance of the Decision Transformer (DT) in offline reinforcement learning. The DT lacks the stitching ability which leads them to learn the sub-optimal trajectories. The authors address this limitation by introducing the counterfactual reasoning module to generate and filter the counterfactual trajectories with high accumulated return and high prediction confidence. The empirical study shows that the proposed method outperforms the existing DT methods in the D4RL benchmark and Atari games.

**Strengths:**

- The CRDT framework aims to address the limitations of the stitching ability of DT by leveraging counterfactual reasoning to generate and filter the counterfactual trajectories without changing the DT's architecture.
- The methodology is inspired by two criteria: high accumulated return and high prediction confidence, which ensures the generated counterfactual trajectories are meaningful and beneficial for the DT's training.
- The empirical results show that the proposed method outperforms the existing DT methods in the D4RL benchmark and Atari games, demonstrating the effectiveness of the CRDT framework.

**Weaknesses:**

- The paper provides a detailed explanation of the CRDT framework but lacks theoretical analysis of the counterfactual reasoning and its impact on the DT's performance.
- The implementation of the CRDT framework involves training two separate transformer models (Treatment and Outcome models), which increases computational consumption and training costs.
- Although this method outperforms existing baseline methods in the average score in the D4RL benchmark and Atari games, the improvements are not significant across all tasks and environments.

**Questions:**

- Can you provide more insights into the counterfactual reasoning module and how the CRDT framework ensures that the generated counterfactual trajectories are beneficial for the DT's training? What is the motivation behind using counterfactual reasoning to improve the DT's performance?
- Since this method uses high accumulated return as the criterion, why are actions leading to the lowest counterfactual returns-to-go chosen?
- Since this method generates counterfactual trajectories of length T/2 in algorithm 2, how does the CRDT framework ensure that these trajectories are meaningful and do not deviate too much from the real trajectories?
- Can you explain why this method performs very well on the medium-expert dataset but has similar performance to the DT on the medium and medium-replay datasets in the D4RL benchmark? What is the reason for this?
- Can you explain why this method may show a significant drop in performance in the Pong game compared to the DT method?

---

> ### Author Response · Authors · 2024-11-21
> **Response to Reviewer ArKg**
>
> Dear Reviewer ArKg,
>
> We deeply appreciate your constructive feedback and the time you’ve dedicated to reviewing our work. We are also grateful for your recognition of our method’s ability to address certain limitations of DT, such as its stitching capability. Moreover, we would like to add that **our primary focus was on** selecting actions in the low-distribution regions of the data as a treatment effect, as a way to **address data sparsity in DT**. We showcase the performance on data sparsity scenarios in Figure 2, wherein our method significantly outperforms existing approaches. We acknowledge that incorporating additional models may increase computational costs during inference compared to the standard DT. However, the training time for both the Outcome and Treatment Networks is approximately 1 hour, thanks to the use of early stopping to limit excessive runtime. The uncertainty threshold for inference is pre-determined through a process that takes around 1 hour. Additionally, generating counterfactual samples requires approximately 2 hours. **Overall, our algorithm's total runtime is 24 hours, compared to 20 hours for DT training and evaluation** under the same conditions. We hope the following responses will help clarify the remaining questions.
>
>
>
> - **Q1** (On Motivation for Incorporating Counterfactual Reasoning): Our primary motivation for incorporating counterfactual reasoning into DT, particularly for selecting actions in the low-distribution regions of the data as treatment effects, is to **address the data sparsity issue**. Our approach is inspired by the findings in [1], which suggests that when improving DT’s performance, the **focus should be on scaling up the dataset** rather than adjusting the architecture. This paper also highlights that DT is more robust than traditional model-based RL methods, such as CQL, when learning with sub-optimal data, allowing us to take a less conservative approach, and select less likely actions in the counterfactual reasoning process. This helps to expand the exploration of action space, especially in underrepresented regions, and improves the model's ability to generalize leading to better performance in the case of data sparsity in Figure 2 and modifying testing environment in Appendix F, Table 4.
>
> [1] Prajjwal Bhargava, Rohan Chitnis, Alborz Geramifard, Shagun Sodhani, and Amy Zhang. When should we prefer decision transformers for offline reinforcement learning? International Conference on Learning Representations, 2024.
>
> - **Q2** (On Choosing Lower Returrns-to-go): To clarify, in our method, lower returns-to-go values correspond to higher accumulated returns because the returns-to-go is a depreciating value, and **a lower value implies that the action taken leads to higher immediate returns**. We choose actions that lead to the lowest counterfactual returns-to-go because these actions are expected to maximize the accumulated return in the future.
>
>
> - **Q3** (On Trajectories Deviating from Sources): We share the concern about counterfactual trajectories deviating too much from the original data. To address this, \textbf{we have incorporated the Counterfactual Action Filtering in our framework} that computes the accumulating uncertainty during the counterfactual reasoning process. This mechanism filters out trajectories that deviate excessively from the original data. In this way, the CRDT framework ensures that the generated counterfactual trajectories are both beneficial (higher returns) and diverse (low-distribution actions), but only when the model is confident in the predictions, as determined by the uncertainty measurement.
>
> - **Q4**: Unfortunately, we politely disagree with this observation, as **CRDT also improves DT baselines on the walker2d-med, walker2d-med-rep, and halfcheetah-med-rep datasets**. However, we do agree that CRDT more consistently achieves state-of-the-art performance on the med-exp dataset. For environments like halfcheetah-med and hopper-med, where performance is similar to DT, this may be due to the specific action distributions in these datasets. We hypothesize that increasing the number of search actions $n_a$ could enhance performance by exploring a broader action space, but this would come at the cost of higher computational complexity. In our paper, we tune $n_a$ between 3 values 3, 5, and 7 and the report results for these two dataset is with $n_a=7$
>
> - **Q5**: Despite the lower performance on Pong, we included it in our manuscript for completeness. We hypothesize that it may be linked to the uncertainty threshold selection process. Our observations suggest that Pong exhibits the highest variability in uncertainty predictions from the Outcome network, potentially related to the input observations.
>
> We hope the clarification will encourage the reviewer to reevaluate and improve your score.

---

> > ### Author Response · Authors · 2024-11-27
> > **Following up with Reviewer ArKg**
> >
> > Dear Reviewer ArKg,
> >
> > With the author-reviewer discussion period nearing its end, we wanted to check if our rebuttal has effectively addressed your concerns. Please let us know if anything remains unclear or if further clarification is needed. Thank again for your thoughtful feedback.

---

### Official Review · Reviewer_PjC8 · 2024-11-06

**Soundness:** 2
**Presentation:** 2
**Contribution:** 2
**Rating:** 5
**Confidence:** 3

**Summary:**

This paper proposes the Counterfactual Reasoning Decision Transformer (CRDT), an extension of the Decision Transformer inspired by the potential outcomes framework from causal inference. CRDT includes a treatment model and an outcome model to enable counterfactual reasoning, allowing the agent to consider hypothetical actions and their potential outcomes. Experiments on locomotion, ant tasks demonstrate that CRDT outperforms DT and traditional methods.

**Strengths:**

1. The paper identifies the limitation of DT that DTs can underperform when optimal trajectories are scarce or data is biased toward suboptimal trajectories.
2. CRDT enables the agent to reason beyond known data by generating counterfactual experiences by integrating the causal inference framework, particularly the potential outcomes approach, with reinforcement learning. CRDT shows better performance on standard benchmarks as compare to traditional methods and DT.

**Weaknesses:**

1. The CRDT framework introduces several new hyperparameters (e.g., number of counterfactual actions, uncertainty threshold, and the number of experiences), which may require extensive tuning for different environments, potentially hindering practical applicability.
2. It is unclear to me how the treatment and outcome models are utilized during inference at test time, as the sections before the experiment sections focus primarily on how to train them. Providing detailed explanations of the inference process (e.g.  through an algo box) would help readers understand the agent's operation in practice.
3. The paper states: "In our framework, the model o is trained with dropout regularization layers. This allows us to run multiple forward passes during the inferencing process to check the uncertainty of the output." Typically, dropout is disabled during inference. The logic behind using dropout during inference for uncertainty estimation needs clarification.
4. Introducing additional models may increase computational costs during inference compared to the standard DT, and the paper lacks a comparison of computational efficiency (e.g., FLOPs, inference time) between CRDT and DT.
5. A demonstration or visualization of what kind of counterfactual actions or states are encountered will help reader understand better how these counterfactual reasoning leads to better performance.

**Questions:**

please see weakness.

---

> ### Author Response · Authors · 2024-11-21
> **Response to Reviewer PjC8**
>
> Dear Reviewer PjC8,
>
> We appreciate your constructive feedbacks and your time reading my reply. We also appreciate your acknowledgment regarding our method being able to tackle certain limitations of DT, for instance, when the data is scares and when data is bias toward suboptimal trajectories.  I hope that the following replies will help clarify your concerns.
>
> - **W1** (On our method having many components): The CRDT framework introduces some additional hyperparameters, but the total number that required tuning is not excessive and is carefully designed to address both continuous and discrete action spaces. In continuous action spaces, the framework requires **three key hyperparameters**: the number of counterfactual actions ($n_a$), which limits the search space for alternative actions; the uncertainty threshold ($\alpha$), which determines the confidence level for counterfactual reasoning; and the number of experiences ($n_e$), which specifies how many past experiences are used for reasoning. These are essential to manage the complexity of the counterfactual search space. In discrete action spaces, only **two** are needed: $\alpha$ and ($n_e$). Compared to other RL or DT frameworks like EDT, which requires multiple loss coefficients and step size parameters, we believe that CRDT's hyperparameters are streamlined and practical.
>
> - **W2** (On training and evaluation process): We understand that the inference process during the test time of the Treatment and Outcome Models might have seemed unclear in the manuscript. The Treatment and the Outcome Networks are not utilized during the training and evaluation of the underlying agent. Instead, in referring to Figure 1 (c), both Treatment and Outcome Network were trained and used to generate counterfactual samples prior to the training of the underlying DT agent. The generated counterfactual samples are then added to a $D_\text{crdt}$ buffer. During the training and evaluation, the underlying agent will sample equally from both the environment dataset and buffer $D_\text{crdt}$ as documented in our **Algorithm Box 3 in Appendix E**. We hope the above points can clarify the process of the framework and have added it to line 199-202 of the manuscript.
>
>
> - **W3** (On dropout uncertainty measurement): We want to clarify that the Outcome Network is used in two different inference processes. **In the first process**, when determining the uncertainty threshold ($\alpha$) for each environment, **the dropout layer is intentionally kept active**. This allows the model's output to vary across multiple forward passes for the same input, enabling us to estimate uncertainty in the predictions. This approach, often referred to as Monte Carlo (MC) Dropout, is a common alternative to ensemble methods, as it provides uncertainty estimates without the computational overhead of training multiple models [1,2]. **In the second process**, when generating the final counterfactual outputs, **the dropout layer is disabled** to ensure the most accurate and deterministic predictions. We have clarified in line 300-303 and line 319 as accordingly.
>
> [1]  Antonio Loquercio, Mattia Segu, and Davide Scaramuzza. "A general framework for uncertainty estimation in deep learning." IEEE Robotics and Automation Letters 5, no. 2 (2020): 3153-3160.
>
> [2] Daily Milanés-Hermosilla, Rafael Codorniú  Trujillo, René López-Baracaldo, Roberto Sagaró-Zamora, Denis Delisle-Rodriguez, John Jairo Villarejo-Mayor, and José Ricardo Núñez-Álvarez. "Monte Carlo dropout for uncertainty estimation and motor imagery classification." Sensors 21, no. 21 (2021): 7241.
>
> - **W4** (On computational cost): We acknowledge that introducing additional models could increase computational costs during inference compared to the standard DT. However, we note that the **training time for both the Outcome and Treatment Networks is approximately 1 hour**. This is achieved using early stopping to prevent excessive time accumulation. **We pre-run the inferencing process to determine the uncertainty threshold for approximately 1 hour**. The time used to generate counterfactual samples is roughly 2 hours. The total time used to run our algorithm is 24 hours compared to 20 hours of DT training and evaluation in the same environment.
>
> - **W5** (On visualization): A visualization of the states in X-Y coordinates for our toy environments is already presented in Figure 1 of the manuscript. In response to your feedback, we will also include an additional figure that illustrates the differences between the distribution of counterfactual actions and original actions in **Figure F.3 and F.4 Appendix F.10**. We also added line 369 in the manuscript to refer to these figures.
>
> We hope the explanation will prompt the reviewer to reconsider and raise your score.

---

> > ### Author Response · Authors · 2024-11-27
> > **Following up with Reviewer PjC8**
> >
> > Dear Reviewer PjC8,
> >
> > Thank you again for your insightful review and feedback. As the author-reviewer discussion period is nearing its conclusion, we’d like to confirm whether our rebuttal sufficiently addresses your concerns. If there’s anything unclear or if further clarification is needed, please feel free to let us know!

---

### Official Review · Reviewer_9nhd · 2024-11-12

**Soundness:** 2
**Presentation:** 3
**Contribution:** 2
**Rating:** 3
**Confidence:** 3

**Summary:**

The authors introduce a Counterfactual Reasoning Decision Transformer (CRDT) that integrates counterfactual actions into the training process.

To create a counterfactual dataset, they select actions based on heuristic criteria—such as a selection probability threshold (e.g., less than $\delta$) and expected ranges for continuous actions.

They then generate trajectories by executing these counterfactual actions, focusing on less confident, low-return-to-go trajectories for the counterfactual dataset.
The Decision Transformer (DT) is trained using both this counterfactual dataset and the original offline dataset.

**Strengths:**

Novel Concept: To the best of my knowledge, this is the first attempt to incorporate counterfactual reasoning into DT training, which adds a novel approach to decision transformers.

**Weaknesses:**

- Weak Connection to Causal Inference: Although inspired by causality, the method lacks a clear link to causal inference concepts, something like causal effects. The approach mainly introduces how to generate uncertain trajectories and incorporate them into DT training.

- Marginal Novelty: The proposed method does not make a strong theoretical contribution beyond its counterfactual data generation process. And the proposed method is not novel.

- Ground-Truth Validity of Counterfactual Trajectories: There’s no clear guarantee that the generated counterfactual trajectories are really real within the ground-truth Markov Decision Process (MDP). This raises concerns about the reliability of using these trajectories in training.

**Questions:**

How to count the number of the encountered input (h_t, s_t+1, g_t+1)?

N_enc is usually 1?

The expected range in (8) is not clear, could any negative value of an action qualify as a counterfactual action, or are there specific restrictions?

The process for selecting candidate counterfactual actions in Lines 271–287 is not fully explained. Could you provide a more detailed description of how these actions are chosen?

---

> ### Author Response · Authors · 2024-11-21
> **Response to Reviewer 9nhd (1/2)**
>
> Dear Reviewer  9nhd,
>
> We sincerely appreciate your constructive feedback and the time you have dedicated to reviewing our work. We are also grateful for your acknowledgment that our method represents the first attempt to incorporate counterfactual reasoning with DT, highlighting its novelty. I hope that my responses address your concerns and provide the necessary clarification.
>
> **Weaknesses**
>
>
>  -  **W1** (On connection to causality): We acknowledge that our method does not fully establish a formal causal structure learning, such as constructing a Structural Causal Model (SCM). However, our work leverages several concepts and studies from causal inference. First, akin to the Potential Outcome framework and research in adapting machine learning methods for causal effect inference, **we use a neural network to estimate the conditional mean function that maps treatment to outcome**. Secondly, while we do not construct an explicit SCM mapping causal relationships between variables, **the attention scores in the Transformer architecture can be interpreted as a simple causal masking mechanism**. Similar methodologies have been used in prior work to determine the causal relationships between inputs and outputs [1,2]. Finally, our framework follows the three key causal assumptions, namely Consistency, Sequential Overlap, and Sequential Ignorability, as detailed in the Appendix. B. We have added a section documenting the above points in the Appendix. G of the revised version and a sentence referring to the Appendix in line 529 of the manuscript.
>
> [1] Uri Shalit, Fredrik D Johansson, and David Sontag. Estimating individual treatment effect: general-
> ization bounds and algorithms. In International conference on machine learning, pp. 3076–3085.
> PMLR, 2017.
>
> [2] Daniel Jacob. Cate meets ml: Conditional average treatment effect and machine learning. Digital
> Finance, 3(2):99–148, 2021.
>
> [3] Silviu Pitis, Elliot Creager, and Animesh Garg. Counterfactual data augmentation using locally
> factored dynamics. Advances in Neural Information Processing Systems, 33:3976–3990, 2020.
>
> [4] Maximilian Seitzer, Bernhard Sch¨olkopf, and Georg Martius. Causal influence detection for improv-
> ing efficiency in reinforcement learning. Advances in Neural Information Processing Systems, 34:
> 22905–22918, 2021.
>
>
>   -  **W2** (On the novelty): We respectfully address the concern regarding the novelty of our proposed method. Firstly, **our work is the first work to integrate counterfactual reasoning into DT training**, providing a novel perspective for tackling data sparsity and enhancing the model’s robustness to suboptimal datasets. Furthermore, our approach aligns with recent insights such as those highlighted in [5], which emphasize the **importance of improving the dataset quality over architectural changes when enhancing DT performance**. By generating diverse counterfactual trajectories and leveraging low-distribution actions effectively, we address this recommendation while extending DT’s ability to generalize. Finally, **the empirical results underscore the practical value of our method**. While we agree that there is room to further refine the theoretical underpinnings, the combination of first-of-its-kind methodology and strong empirical evidence establishes the novelty and impact of our contribution.
>
> [5] Prajjwal Bhargava, Rohan Chitnis, Alborz Geramifard,Shagun Sodhani, and Amy Zhang. When
> should we prefer decision transformers for offline reinforcement learning? International Conference on Learning Representations, 2024.
>
>    -  **W3** (On unrealistic trajectories): **The trajectories generated by our method rely on the Outcome and Treatment Networks, which are trained on real trajectories**. This training anchors the counterfactual trajectories to the underlying data distribution, reducing the likelihood of unrealistic trajectories. We also incorporate an uncertainty measurement mechanism to filter out trajectories that deviate excessively from the data distribution. By focusing on predictions with low uncertainty, we ensure the generated trajectories remain within the plausible bounds of the MDP.

---

> ### Author Response · Authors · 2024-11-21
> **Response to Reviewer 9nhd (2/2)**
>
> **Questions**
>
>   -  **Q1**: During the training of the Treatment Network, occurrences of input tuples (h\_t, s\_{t+1}, g\_{t+1}) are tracked using a hashing function to map each input into a hash table, where the counts are stored. These counts are later used during the counterfactual reasoning process. This approach is inspired by research on count-based exploration methods, where the frequency of encountered inputs is similarly recorded [6].
>
> [6] Tang Haoran, Rein Houthooft, Davis Foote, Adam Stooke, OpenAI Xi Chen, Yan Duan, John Schulman, Filip DeTurck, and Pieter Abbeel. "\# exploration: A study of count-based exploration for deep reinforcement learning." Advances in neural information processing systems 30, 2017.
>
>    - **Q2**: $n_{enc}$  is often 1 from the 2nd counterfactual generation step onward as the history h\_t is not met during training.
>
>    -  **Q3**: In our work, all treatments that are modified and result in better returns are selected as counterfactual treatments. Our method seeks to maximize the utility of the dataset, even when it is suboptimal, by choosing actions that are less likely to be selected but have the potential to yield higher returns. Crucially, these actions are only considered if the model demonstrates confidence in its predictions, meaning the uncertainty associated with these actions is low.
>
>   - **Q4**: We understand that the action selection process of the Treatment and Outcome Models might have seemed unclear in the manuscript. To address this, **we have included Algorithm Box 2 in Appendix D** (we omit it from the main paper due to page limitation), which provides a more detailed explanation of the inference process. To elaborate, we first compute a set of modified treatments ($n_a$ is the number of actions that we will try) using Eq.9 from the manuscript (where the step size for each search range is $\beta$). Each modified treatment replaces the original treatment and is fed into the Outcome Model, resulting in a new trajectory and updated returns-to-go at the current step. Among these trajectories, the one associated with the lowest returns-to-go, at the current step, and does not result in an accumulated uncertainty higher than the predefined threshold is selected as the counterfactual trajectory.
>
> We hope the clarification will lead the reviewer to increase your score. Thank you for your time!

---

> > ### Author Response · Authors · 2024-11-27
> > **Following up with Reviewer 9nhd**
> >
> > Dear Reviewer 9nhd,
> >
> > As the author-reviewer discussion period is coming to a close, we wanted to check if our rebuttal has adequately addressed your concerns. Please don’t hesitate to let us know if anything remains unclear or requires further clarification!

---

### Author Response · Authors · 2024-11-21
**General response**

We are grateful to all the reviewers for their detailed and insightful feedback on our submission.
To briefly summarize the reviews: Reviewers reported that our paper aims to address an important problem and limitation of the DT method (Reviewers 9nhd, PjC8, ArKg, cDB6) and we are the first that try to incorporate counterfactual reasoning into DT training, which “added a novel approach to DT” (Reviewer 9nhd). Moreover, empirical results show that CRDT outperform on standard benchmarks as compared to traditional methods and DT (Reviewers PjC8, ArKg).
The rebuttal has been organized in terms of individual responses to each reviewer. We have also submitted a pdf revision of the manuscript, with the following updates:

-	Appendix G about relation to causal inference and counterfactual reasoning.
-	A summary of the flow of our framework in line 199-202.
-	Visualization of action distribution in Figure F.3 and F.4 Appendix F.10.
-	Empirical results on random dataset in Table 9 of Appendix F.9.

Should any additional questions arise, or if further clarification is needed, we will continue clarifying on this page throughout the discussion period.

---

### Meta-Review · Area_Chair_bU1t · 2024-12-20

**Metareview:**

This paper introduces Counterfactual Reasoning Decision Transformer to enhances DT's ability to reason in the sparse data and out-of-distribution scenarios. It achieves this by generating counterfactual experiences.

While reviewers acknowledge the importance of the sparse-data problem for DT and its novelty of incorporating counterfactual reasoning as a promising direction, all reviewers are concerned on the effectiveness of the proposed method. Particularly, they share concerns on the lack of theoretical connection to causal inference, the validity of the counterfactual reasoning, additional computational cost, and insufficient empirical evaluations.

**Additional Comments On Reviewer Discussion:**

Authors' feedback is not sufficient to address the main concerns.

---

### Decision · Program_Chairs · 2025-01-22

Reject